# StableGuard: Towards Unified Copyright Protection and Tamper Localization in Latent Diffusion Models

**Haoxin Yang**[1]    **Bangzhen Liu**[1]    **Xuemiao Xu**[1†]    **Cheng Xu**[2†]    **Yuyang Yu**[1]
**Zikai Huang**[1]    **Yi Wang**[3]    **Shengfeng He**[2]

[1]South China University of Technology    [2]Singapore Management University
[3]Dongguan University of Technology

harxis@outlook.com    xuemx@scut.edu.cn
{liubz.scut, cschengxu, yyyoung0611, zikaihuang0428}@gmail.com
yiwang@dgut.edu.cn    shengfenghe@smu.edu.sg
https://github.com/Harxis/StableGuard

## Abstract

The advancement of diffusion models has enhanced the realism of AI-generated content but also raised concerns about misuse, necessitating robust copyright protection and tampering localization. Although recent methods have made progress toward unified solutions, their reliance on post hoc processing introduces considerable application inconvenience and compromises forensic reliability. We propose StableGuard, a novel framework that seamlessly integrates a binary watermark into the diffusion generation process, ensuring copyright protection and tampering localization in Latent Diffusion Models through an end-to-end design. We develop a Multiplexing Watermark VAE (MPW-VAE) by equipping a pretrained Variational Autoencoder (VAE) with a lightweight latent residual-based adapter, enabling the generation of paired watermarked and watermark-free images. These pairs, fused via random masks, create a diverse dataset for training a tampering-agnostic forensic network. To further enhance forensic synergy, we introduce a Mixture-of-Experts Guided Forensic Network (MoE-GFN) that dynamically integrates holistic watermark patterns, local tampering traces, and frequency-domain cues for precise watermark verification and tampered region detection. The MPW-VAE and MoE-GFN are jointly optimized in a self-supervised, end-to-end manner, fostering a reciprocal training between watermark embedding and forensic accuracy. Extensive experiments demonstrate that StableGuard consistently outperforms state-of-the-art methods in image fidelity, watermark verification, and tampering localization.

## 1 Introduction

The unprecedented capabilities of latent diffusion models (LDMs) have revolutionized image synthesis, redefining artistic creation and visual content production [1–6]. However, as these models become more accessible, concerns over unauthorized use and malicious content tampering have intensified, necessitating solutions to safeguard intellectual property and ensure content authenticity.

Traditional copyright protection relies on watermarking techniques [7–9], which embed identifiable information into images through post-hoc processing. While effective in some cases, these methods introduce computational overhead and image degradation due to their detachment from the generative process. Recent approaches to diffusion-native watermarking integrate embedding mechanisms directly within the generation pipeline [10–14], but they fail to support advanced forensic needs

---

[†] Corresponding authors.

39th Conference on Neural Information Processing Systems (NeurIPS 2025).

such as quantifying tampering severity or precisely localizing manipulations [15–20]. Although recent works have attempted to unify copyright protection and tampering localization in a single framework [21–23], they remain inherently post-hoc, with generation and forensics optimized independently. This decoupled optimization prevents mutual enhancement between the two tasks, thereby limiting both the overall forensic robustness and practical applicability. The absence of a seamless framework that integrates these functionalities in an LDM-native manner remains a major gap, hindering real-world deployment in security-sensitive applications.

To bridge the gap between copyright protection and content forensics in LDMs, we propose *StableGuard*, a unified framework that integrates holistic watermark directly into the generative process. Our core insight is twofold: (i) holistically distributed watermarks exhibit inherent robustness against localized manipulations due to spatial redundancy, and (ii) the subtle perturbations introduced by these watermarks can serve as reliable cues for tampering localization through missing feature detection. These complementary properties make holistic watermarking a compelling foundation for jointly addressing ownership verification and fine-grained forensic analysis in a unified LDM-native manner. StableGuard achieves this integration through two key components: the *Multiplexing Watermark Variational Autoencoder (MPW-VAE)* for embedding watermarks during generation, and the *Mixture-of-Experts Guided Forensic Network (MoE-GFN)* for tamper-agnostic forensic analysis. MPW-VAE extends the standard LDM architecture by incorporating a trainable watermark adapter within the VAE decoder and employs a multiplexing strategy to generate both visually indistinguishable watermarked and watermark-free images, facilitating self-supervised training

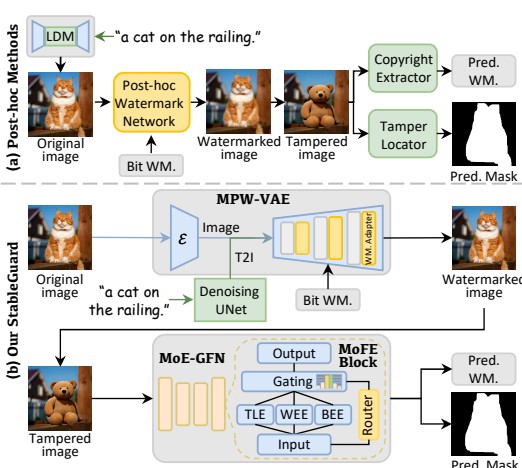

Figure 1: (a) Unlike other post-hoc methods that add external watermarking, introducing overhead and detaching from the generative process. (b) StableGuard embeds a holistic binary watermark directly into the diffusion pipeline and adaptively extracts intrinsic perturbation patterns via collaborative forensic experts, enabling unified, precise, and seamless protection within LDMs.

without manual annotations. MoE-GFN performs watermark extraction and tampering localization via a *Mixture-of-Forensic-Experts (MoFE)* block comprising three specialized experts: a *Watermark Extraction Expert*, a *Tampering Localization Expert*, and a *Boundary Enhancement Expert*. These experts are dynamically fused by a *Dynamic Soft Router*, allowing for precise and holistic forensic predictions. The MPW-VAE and MoE-GFN are trained jointly in an end-to-end manner, enabling mutual reinforcement between watermark embedding and forensic performance. Extensive experiments across multiple benchmarks demonstrate that StableGuard outperforms existing methods in both watermark retrieval and tampering localization, validating its effectiveness for real-world security-sensitive applications.

In summary, our main contributions are as follows:

- We introduce StableGuard, a unified proactive forensics framework for LDMs, seamlessly integrating copyright protection and tampering localization into LDM-based image generation.

- We propose a simple yet effective self-supervised forensic learning framework that embeds an imperceptible, holistic bit watermark via a multiplexing watermark VAE. This enables precise copyright protection and tampering localization without requiring labeled data.

- We develop a tampering-agnostic mixture-of-experts forensic network that synergistically integrates holistic, subtle, and boundary features to extract forensic cues, ensuring robust watermark retrieval and precise tampering localization across various attack scenarios.

- Extensive experiments demonstrate that StableGuard outperforms state-of-the-art methods in both forensic accuracy and robustness while preserving the visual fidelity of generated images.

## 2 Related Work

### 2.1 Image Watermark

Image watermarking has long served as a foundational technique for intellectual property protection [24–28]. Traditional approaches typically rely on encoder and decoder architectures [7, 8], flow-based models [21, 29], or per-image encoding strategies [7, 30]. These methods are often implemented as separate steganographic modules applied after image generation, introducing additional processing overhead and potential degradation in visual quality. In parallel, diffusion models, particularly latent diffusion models (LDMs), have recently achieved remarkable progress across diverse domains [1, 2, 31–35]. Motivated by this success, recent studies have explored embedding watermarks directly into the generative process [10–12, 14, 36]. For example, Tree-Ring [36] modifies the initial noise vector, CRoSS [14] employs image-to-image steganography, and methods such as Stable Signature [10], WOUAF [11], and WaDiff [12] fine-tune LDMs to embed watermarks in latent space. While these LDM-native approaches improve integration and visual fidelity, they generally neglect tampering localization—a key aspect for assessing manipulation severity and enabling forensic content reuse. Our work introduces a unified framework that jointly addresses copyright protection and tampering localization in LDMs, leveraging mutual optimization between watermark and forensic analysis to deliver robust, high-fidelity image forensics.

### 2.2 Tampering Localization

Tampering localization aims to detect and localize manipulated regions, crucial for verifying image authenticity and quantifying alterations [15–21, 37, 38]. Existing methods fall into two broad categories: passive detection and proactive defense. Passive methods rely solely on visual artifacts. For instance, MVSSNet [15] captures boundary inconsistencies, IML-ViT [16] extracts multi-scale features via a transformer backbone, and PSCC-Net [17], ObjectFormer [18], and HDF-Net [19] enhance spatial awareness using complementary cues. However, these approaches often require paired supervision and struggle to generalize across manipulation types. Proactive defenses embed auxiliary signals into the image to aid future tamper detection. Traditional fragile watermark offers block-level localization but lacks adaptability [39–43]. Recent methods like MaLP [44], EditGuard [21], OmniGuard [22] and WAM [23] use learned templates or invertible flows to embed watermarks for both copyright and localization. Other approaches [45, 46] explore semi-supervised tampering localization, offering post-hoc forensic analysis without prior watermarking. Despite their advantages, these post-hoc designs require external embedding networks, introduce application complexity, and are prone to quality degradation, especially for LDM-generated content. Moreover, methods like EditGuard and OmniGuard rely on reference images [21, 22], making them sensitive to auxiliary input quality. In contrast, our proposed StableGuard integrates watermark directly into the LDM process, eliminating the need for post-processing. This diffusion-native approach unifies copyright protection and tampering localization within a single, end-to-end framework where both tasks reinforce one another.

## 3 Proposed Method

### 3.1 Preliminaries of Latent Diffusion Models

Latent Diffusion Models (LDMs) are a class of generative models that synthesize images by iterative denoising a latent representation [1–5]. Unlike conventional diffusion models [47, 48] that operate in pixel space, LDMs first encode images into a lower-dimensional latent space, enabling more efficient training and inference while preserving image fidelity. The generative process begins by sampling a latent vector $z_T \sim \mathcal{N}(0, I)$, which is progressively denoised through the recurrence:

$$z_{t-1} = \frac{1}{\sqrt{\alpha_t}} \left( z_t - \frac{1 - \alpha_t}{\sqrt{1 - \bar{\alpha}_t}} \epsilon_\theta(z_t, t) \right), \tag{1}$$

where $z_t$ is the latent at timestep $t$, $\alpha_t$ is the noise schedule, and $\bar{\alpha}_t = \prod_{s=1}^{t} \alpha_s$. The model $\epsilon_\theta$ predicts the added noise at each step. This iterative refinement continues until the clean latent $z_0$ is recovered. Finally, $z_0$ is decoded into image space via a VAE decoder:

$$X = \text{VAE Decoder}(z_0). \tag{2}$$

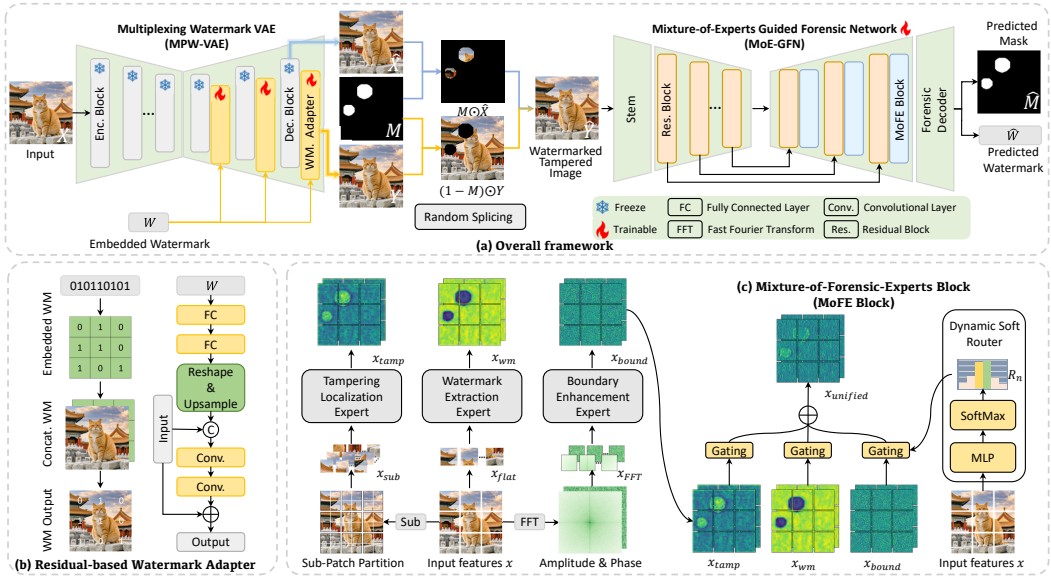

Figure 2: Overview of the StableGuard Framework. StableGuard integrates an MPW-VAE and an MoE-GFN in a self-supervised, end-to-end pipeline. The MPW-VAE generates visually consistent watermarked and non-watermarked image pairs from the same latent code by introducing a lightweight, residual-based watermark adapter into the LDM decoder. These image pairs, augmented via random masking, form a diverse training set for MoE-GFN training. MoE-GFN adaptively fuses features from multiple forensic experts to extract robust, unified signals from holistic watermark perturbations. Joint optimization of both components enables reciprocal enhancement of watermark embedding and forensic analysis, supporting accurate copyright verification and tampering localization.

## 3.2 Overview

To enable imperceptible watermark embedding and robust forensic analysis within LDMs, we propose StableGuard, a unified framework for copyright protection and tampering localization that integrates seamlessly into the LDM pipeline. As illustrated in Fig. 2, StableGuard consists of two core components: the *Multiplexing Watermark VAE (MPW-VAE)* and the *Mixture-of-Experts Guided Forensic Network (MoE-GFN)*. MPW-VAE introduces a hierarchy of lightweight, latent residual-based watermark adapters into the LDM's VAE decoder, enabling holistic binary watermark embedding while preserving perceptual fidelity. Crucially, watermark adapters can be toggled on or off, facilitating the generation of paired watermarked and clean samples from shared latent, supporting fully self-supervised training of the forensic module. For downstream analysis, MoE-GFN leverages the subtle perturbations introduced by the watermark to perform both extraction and tampering localization. Central to this network is the *Mixture-of-Forensic-Experts (MoFE) Block*, comprising three expert branches specialized in capturing complementary forensic cues. Their outputs are adaptively fused through a dynamic soft routing mechanism, allowing the network to selectively emphasize relevant features and achieve robust, accurate forensic predictions.

## 3.3 Multiplexing Watermark VAE

Most existing passive tampering localization methods depend on manually annotated masks for supervised learning, which is not only labor-intensive but also exhibits poor generalization across diverse manipulation scenarios [15–19]. To address these challenges, we propose the MPW-VAE, which incorporates a togglable lightweight watermark injection branch into a pre-trained LDM VAE decoder. This multiplexing design enables the generation of visually indistinguishable watermarked and watermark-free image pairs from the same latent input, providing a scalable foundation for self-supervised learning in forensic tasks.

To preserve visual fidelity, watermark embedding is achieved through latent residual-based adapters inserted after each block of the VAE decoder. Each adapter encodes the watermark via two fully connected layers, reshapes the embedding to match the feature dimensions, and concatenates it with the decoder features. The fused features are then processed through two convolutional blocks, followed by a residual connection to the input. This design ensures coherent and unobtrusive watermark integration (see Fig. 2(b)). Leveraging this residual mechanism, MPW-VAE produces watermarked images with imperceptible yet traceable watermark signals, while maintaining high visual similarity to the watermark-free outputs generated by the original decoder.

To simulate diverse and realistic tampering scenarios, we synthesize training samples by randomly fusing watermarked images $Y$ with their watermark-free counterparts, either the VAE-reconstructed image $\hat{X}$ or the original image $X$, using a random binary mask $M$:

$$\begin{cases} \hat{Y} = (1 - M) \odot Y + M \odot X, & p \le 0.5 \\ \hat{Y} = (1 - M) \odot Y + M \odot \hat{X}, & p > 0.5 \end{cases} \quad (3)$$

where $\hat{Y}$ denotes the synthesized training image, and $p$ controls whether the spliced region is drawn from the VAE reconstruction or the original image. Specifically, we use $X$ (a real image) simulates human-made manipulations produced by editing tools (e.g., Photoshop), while using $\hat{X}$ (the vanilla LDM VAE reconstruction) emulates AI-generated forgeries increasingly common with generative models. To diversify tampering patterns, we adopt a hybrid masking strategy: with 50% probability, we apply random binary masks; otherwise, we use semantic masks drawn from a pool of SAM-generated segmentations [49]. These SAM-based masks are non-image-paired, ensuring shape diversity without extra supervision. This augmentation strategy enables the network to learn to distinguish watermarked from non-watermarked regions in a self-supervised manner, thereby supporting robust joint learning of watermark extraction and tampering localization.

### 3.4 Mixture-of-Experts Guided Forensic Network

The embedding of holistic watermarks into generated images enables not only robust watermark verification but also tampering localization by identifying disruptions in the underlying perturbation patterns. To effectively harness the synergy between these two tasks, we propose the Mixture-of-Experts Guided Forensic Network (MoE-GFN). Inspired by the mixture-of-experts paradigm [50], MoE-GFN integrates task-specialized modules to enhance forensic performance through collaborative feature modeling. As illustrated in Fig. 2(a), MoE-GFN adopts a UNet-based architecture[51], where the decoder is augmented with a Mixture-of-Forensic-Experts (MoFE) Block (Fig. 2(c)). This block comprises three dedicated experts—Watermark Extraction, Tampering Localization, and Boundary Enhancement—each designed to capture complementary forensic cues. Their outputs are fused via a Dynamic Soft Router, which adaptively integrates task-relevant features based on the input. The complete pipeline includes a Stem module[52] for dimensionality reduction, a UNet backbone for multi-scale feature extraction, and a Forensic Decoder for final prediction.

**Watermark Extraction Expert.** To recover globally embedded watermarks even under local modifications, we employ a transformer-based expert that captures long-range dependencies. By modeling global correlations, this module provides robust watermark reconstruction and indirectly facilitates tampering localization by highlighting regions where watermark patterns are missing. Given input features $x \in \mathbb{R}^{b,c,h,w}$, we reshape them to $x_{\text{flat}} \in \mathbb{R}^{b,hw,c}$ and process them through a linear projection (Proj), followed by Multi-Head Self-Attention (MHSA) and a Feed-Forward Network (FFN) [53]. The final output is computed with residual addition:

$$x_{\text{wm}} = \text{FFN}(\text{MHSA}(\text{Proj}(x_{\text{flat}}))) + x_{\text{flat}}. \quad (4)$$

**Tampering Localization Expert.** In contrast to watermark extraction, tampering localization demands sensitivity to subtle, local inconsistencies. To this end, we propose a sub-patch transformer that partitions feature maps into smaller patches and applies localized attention to enhance the detection of fine-grained manipulation artifacts. Formally, the input $x$ is partitioned into $n \times n$ patches and reshaped to $x_{\text{sub}} \in \mathbb{R}^{b \cdot \frac{hw}{n^2}, n^2, c}$. It is then processed as:

$$x_{\text{tamp}} = \text{FFN}(\text{MHSA}(\text{Proj}(x_{\text{sub}}))) + x_{\text{sub}}. \quad (5)$$

**Boundary Enhancement Expert.** To further refine tampering boundaries, we incorporate frequency-domain cues through a Boundary Enhancement Expert, which highlights high-frequency components

indicative of manipulation edges. This expert applies a transformer in the Fourier domain to capture boundary-specific anomalies that may be missed in the spatial domain. Given input $x$, we compute its Fourier transform, reshape the frequency representation, and process it through transformer layers before applying the inverse transform:

$$x_{\text{FFT}} = \text{Reshape}(\text{FFT}(x), (b, hw, c)),$$
$$x_{\text{bound}} = \text{iFFT}(\text{FFN}(\text{MHSA}(\text{Proj}(x_{\text{FFT}}))) + x_{\text{FFT}}). \tag{6}$$

where FFT and iFFT denote the Fourier transform and inverse Fourier transform, respectively.

**Dynamic Soft Router.** To adaptively fuse the outputs of the three expert branches, we introduce a Dynamic Soft Router, which predicts expert-specific fusion weights conditioned on the input. This mechanism enables MoE-GFN to dynamically modulate its focus based on the type and spatial characteristics of the tampering. Given a shared input feature map $x$, the router generates soft weights $R = \text{Softmax}(f(x))$, where $f(\cdot)$ is a lightweight multi-layer perceptron. The unified forensic representation is computed as a weighted sum:

$$x_{\text{unified}} = \sum_{n=1}^{3} R_n \odot \text{Expert}_n(x). \tag{7}$$

By enabling input-adaptive feature integration, the Dynamic Soft Router enhances the model's ability to generalize across diverse manipulation patterns, yielding more resilient forensic predictions.

### 3.5 Loss Function

StableGuard is trained in a fully self-supervised, end-to-end manner, enabling mutual enhancement between watermark embedding and its watermark extraction and tampering localization tasks. During MPW-VAE training, we freeze the parameters of the pre-trained LDM VAE decoder and update only the watermark adapter. This strategy ensures high perceptual fidelity between the original and watermarked images while effectively encoding watermark information. The total training objective comprises three components: a similarity loss to maintain image quality, a watermark loss to ensure accurate extraction, and a tampering loss to localize manipulated regions precisely.

**Similarity Loss.** To preserve visual consistency between the watermarked image $Y$ and the reconstructed watermark-free image $\hat{X}$, we combine L1 distance with perceptual similarity [54]:

$$\mathcal{L}_{\text{sim}} = ||\hat{X} - Y||_1 + \text{PS}(\hat{X}, Y). \tag{8}$$

**Watermark Loss.** We supervise watermark extraction using a binary cross-entropy loss:

$$\mathcal{L}_{\text{wm}} = -\frac{1}{L} \sum_l \left( W_l \cdot \log(\hat{W}_l) + (1 - W_l) \cdot \log(1 - \hat{W}_l) \right), \tag{9}$$

where $W$ and $\hat{W}$ are the ground truth and predicted watermark bits, and $L$ is the total bit length.

**Tampering Loss.** To achieve precise localization of tampered regions, we combine weighted binary cross-entropy (WBCE) with Dice loss [55]:

$$\mathcal{L}_{tamper} = \lambda_0 \mathcal{L}_{\text{wbce}} + (1 - \lambda_0) \mathcal{L}_{\text{dice}}, \tag{10}$$

$$\mathcal{L}_{\text{wbce}} = -\frac{1}{N} \sum_{i,j} \left( \lambda_1 \cdot M_{i,j} \cdot \log(\sigma(\hat{M}_{i,j})) + \lambda_2 \cdot (1 - M_{i,j}) \cdot \log(1 - \sigma(\hat{M}_{i,j})) \right), \tag{11}$$

$$\mathcal{L}_{\text{dice}} = 1 - \frac{2 \sum_{i,j} M_{i,j} \cdot \sigma(\hat{M}_{i,j})}{\sum_{i,j} \left( M_{i,j} \right)^2 + \sum_{i,j} \left( \sigma(\hat{M}_{i,j}) \right)^2}, \tag{12}$$

where $M$ and $\hat{M}$ denote the ground truth and predicted tampering masks, $\sigma$ is the sigmoid function, $N$ is the total number of pixels, $i, j$ are the pixel indices, and $\lambda_1, \lambda_2$ are weighting factors for foreground and background.

**Total Loss.** The final training objective for the StableGuard framework is defined as:

$$\mathcal{L}_{\text{total}} = \mathcal{L}_{\text{sim}} + \mathcal{L}_{\text{wm}} + \mathcal{L}_{\text{tamper}}. \tag{13}$$

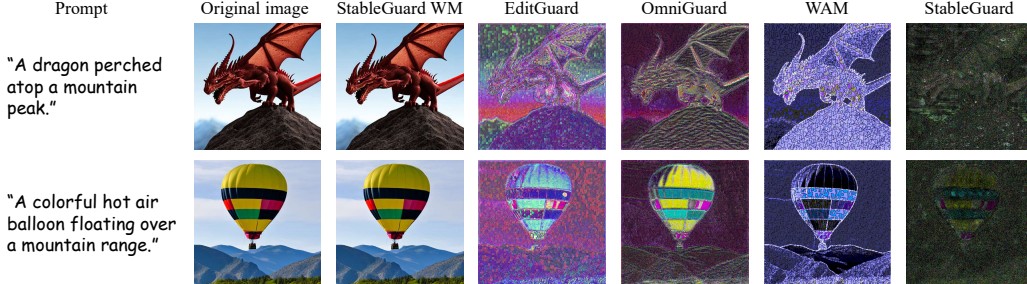

Figure 3: Comparison between the original Stable Diffusion VAE and a variant using our MPW-VAE alongside other watermarking methods [21–23]. To better visualization, pixel-wise differences are amplified $100\times$. StableGuard embeds holistic watermarks that remain visually imperceptible while preserving image quality.

# 4 Experiments

## 4.1 Settings

**Datasets.** We train StableGuard on the COCO training set [56] in a fully self-supervised manner, without requiring any manual tampering annotations. For evaluation, watermark extraction is tested on both the COCO test set and a custom text-to-image (T2I) dataset, measuring performance on real and LDM-generated content. To assess tampering localization, we focus on robustness against AI-generated manipulations using a large-scale AIGC benchmark of 35,000 images—25,000 from COCO and 10,000 from T2I dataset. Semantic masks are generated with SAM [49], followed by region edits using Stable Diffusion [1], SDXL [2], Kandinsky [57], ControlNet [4], and LaMa [58]. *Further details are provided in the Appendices.*

**Implementation Details.** StableGuard is implemented in PyTorch, built upon Stable Diffusion 2.1[1]. During training, we freeze the base model and optimize only the Watermark Adapter and MoE-GFN using the Adam optimizer [59] with a learning rate of $1 \times 10^{-4}$. All experiments are run for 10 epochs on two NVIDIA RTX 4090D GPUs with a batch size of 16. The tampering localization expert operates on sub-patches of size $n = 8$. Loss weights are set as $\lambda_0 = 0.2$ (Eq. (10)), $\lambda_1 = 2$, and $\lambda_2 = 0.5$ (Eq. (11)).

## 4.2 Comparison on Watermarking

The effectiveness of our proposed StableGuard for watermark embedding and extraction is validated by experiments on two datasets: the COCO dataset and the T2I dataset. To intuitively demonstrate StableGuard's watermark injection capability, we showcase two watermark embedding examples from the T2I dataset and compare with three post-hoc watermarking methods [21–23] in Fig. 3. The corresponding prompts for generating images are *"A dragon perched atop a mountain peak"* and *"A colorful hot air balloon floating over a mountain range"*, respectively. As observed, the images before and after watermark embedding exhibit high visual consistency, with pixel-wise differences at a negligible level. The difference maps reveal globally distributed pixels across the entire image, demonstrating that StableGuard indeed effectively embeds holistic imperceptible watermarks while maintaining high-fidelity image generation results.

We also quantitatively compare the performance of StableGuard with eight state-of-the-art watermarking techniques under different watermark bit lengths (B.L.). The compared methods include five post-hoc watermarking approaches: HiDDeN [7], SepMark [8], EditGuard [21], OmniGuard [22] and WAM [23], as well as three diffusion-based methods: Stable Signature [10], WOUAF [11], and WaDiff [12]. As shown in Tab. 1, our approach maintains competitive with existing solutions on effectively preserving the perceptual quality of the original content after watermark embedding, where the embedded images exhibit close visual similarity with their non-watermarked counterparts.

---

[1] https://huggingface.co/stabilityai/stable-diffusion-2-1-base

Table 1: Quantitative comparison on COCO [56] and the T2I dataset. The best and second-best results are **bolded** and underlined, respectively.

| Method | B.L. | COCO Dataset [56] | | | | | T2I-Dataset | | | | |
|---|---|---|---|---|---|---|---|---|---|---|---|
| | | PSNR↑ | SSIM↑ | LPIPS↓ | FID↓ | Bit Acc↑ | PSNR↑ | SSIM↑ | LPIPS↓ | FID↓ | Bit Acc↑ |
| HiDDeN [7] | 32 | 31.95 | 0.879 | 0.104 | 20.0 | 98.80 | 32.03 | 0.881 | 0.101 | 19.9 | 98.79 |
| SepMark [8] | 32 | 35.38 | 0.929 | 0.090 | 20.3 | 98.85 | 35.42 | 0.931 | 0.089 | 20.1 | 98.86 |
| WOUAF [11] | 32 | 31.20 | 0.906 | 0.115 | 21.6 | 99.13 | 31.14 | 0.908 | 0.113 | 21.7 | 99.14 |
| Stable Signature [10] | 48 | 30.12 | 0.890 | 0.120 | 20.9 | 99.12 | 30.06 | 0.892 | 0.118 | 20.8 | 99.10 |
| WaDiff [12] | 48 | 35.55 | 0.960 | 0.085 | 19.8 | 98.14 | 35.49 | 0.962 | 0.083 | 19.6 | 98.13 |
| EditGuard [21] | 64 | 32.75 | 0.937 | 0.105 | 20.0 | 99.77 | 32.87 | 0.939 | 0.103 | 19.9 | 99.78 |
| OmniGuard [22] | 100 | 37.54 | 0.950 | 0.072 | 20.1 | 98.11 | 37.32 | 0.944 | 0.070 | 20.0 | 98.09 |
| WAM [23] | 32 | 38.20 | 0.951 | 0.067 | 19.9 | 98.17 | 37.80 | 0.946 | 0.073 | 19.6 | 98.33 |
| Ours | 32 | **40.50** | **0.970** | **0.062** | **19.5** | **99.97** | **40.53** | **0.972** | **0.060** | **19.4** | **99.98** |
| | 48 | 40.42 | 0.969 | 0.063 | 19.7 | 99.96 | 40.47 | 0.971 | 0.061 | 19.6 | **99.98** |
| | 64 | 40.40 | 0.968 | 0.065 | 19.8 | 99.95 | 40.44 | 0.970 | 0.063 | 19.7 | 99.96 |
| | 128 | 40.10 | 0.966 | 0.070 | 19.9 | 99.87 | 40.11 | 0.968 | 0.069 | 19.8 | 99.88 |
| | 256 | 40.05 | 0.964 | 0.073 | 20.1 | 99.83 | 40.08 | 0.965 | 0.062 | 20.0 | 99.84 |

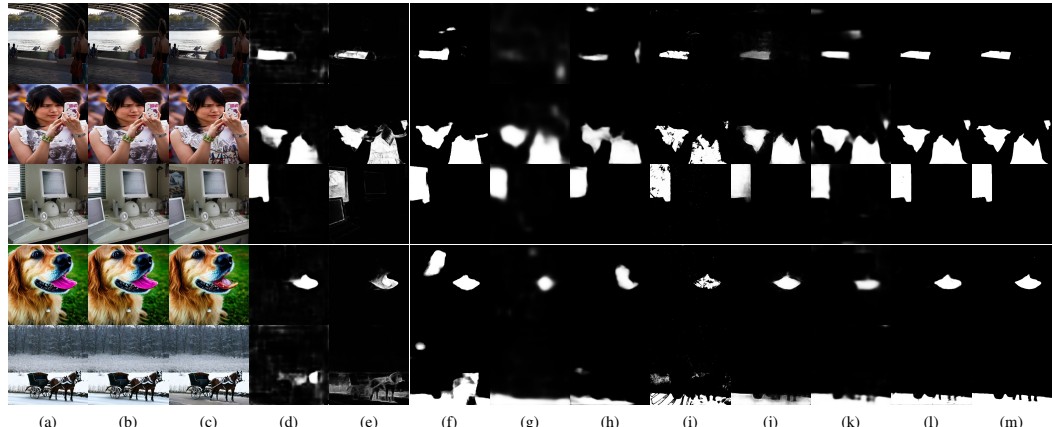

| (a) | (b) | (c) | (d) | (e) | (f) | (g) | (h) | (i) | (j) | (k) | (l) | (m) |

Figure 4: The visualization of tampering localization results on the AIGC tampering dataset. (a) is the original image, (b) is the watermarked image by StableGuard, (c) is the tampered image, (d)-(m) are the tampering localization results of MVSS-Net [15], IML-ViT [16], PSCC-Net [17], ObjectFormer [18], HDF-Net [19], EditGuard [21], OmniGuard [22], WAM [23], our StableGuard, and the ground truth, respectively. Please zoom in for best view.

Meanwhile, the accuracy of the watermark extracted from these images consistently outperforms existing techniques.

Unlike conventional post-hoc approaches, our method is seamlessly integrated into the LDM generation pipeline, exploiting an end-to-end model to directly embed watermarks into generated images. Furthermore, compared to those diffusion-native watermarking methods, our tailored MPW-VAE employs a latent residual-based watermark adapter after the VAE decoder block, which is jointly trained with the forensics tasks, thus achieving adaptive holistic watermark injection while minimally altering the original image content.

## 4.3 Comparison on Tampering Localization

The quantitative results of various methods on the AIGC tampering dataset are presented in Tab. 2. As shown, existing passive localization techniques exhibit notably inferior performance when compared to StableGuard, highlighting their inability to effectively detect the latest AIGC tampering methods. In contrast, StableGuard demonstrates robust detection capabilities across a range of AIGC tampering techniques, validating the effectiveness of our approach. When compared to EditGuard, OmniGuard and WAM, StableGuard outperforms them on most metrics. This improvement is a direct result of StableGuard's design, which integrates the unique characteristics of both watermarking and tampering detection. Unlike other methods, which identify tampered regions by detecting anomalies

Table 2: Localization precision comparison on the AIGC tampering dataset.

| Method | SD Inpainting [1] | | | SD XL [2] | | | Kandinsky [57] | | | ControlNet [4] | | | LAMA [58] | | |
|---|---|---|---|---|---|---|---|---|---|---|---|---|---|---|---|
| | F1↑ | AUC↑ | IoU↑ | F1↑ | AUC↑ | IoU↑ | F1↑ | AUC↑ | IoU↑ | F1↑ | AUC↑ | IoU↑ | F1↑ | AUC↑ | IoU↑ |
| MVSS-Net [15] | 0.862 | 0.934 | 0.791 | 0.848 | 0.929 | 0.775 | 0.848 | 0.928 | 0.775 | 0.856 | 0.930 | 0.782 | 0.860 | 0.934 | 0.789 |
| IML-ViT [16] | 0.907 | 0.923 | 0.879 | 0.904 | 0.921 | 0.876 | 0.906 | 0.923 | 0.877 | 0.898 | 0.883 | 0.840 | 0.898 | 0.894 | 0.880 |
| PSCC-Net [17] | 0.898 | 0.976 | 0.829 | 0.899 | 0.977 | 0.830 | 0.894 | 0.975 | 0.825 | 0.899 | 0.977 | 0.830 | 0.898 | 0.976 | 0.829 |
| ObjectFormer [18] | 0.476 | 0.722 | 0.398 | 0.479 | 0.719 | 0.398 | 0.472 | 0.718 | 0.398 | 0.467 | 0.724 | 0.390 | 0.503 | 0.738 | 0.425 |
| HDF-Net [19] | 0.556 | 0.762 | 0.468 | 0.763 | 0.470 | 0.560 | 0.544 | 0.759 | 0.457 | 0.551 | 0.764 | 0.463 | 0.565 | 0.767 | 0.476 |
| EditGuard [21] | 0.937 | 0.977 | 0.911 | 0.938 | 0.976 | 0.913 | 0.935 | 0.966 | 0.913 | 0.939 | 0.969 | 0.907 | 0.939 | 0.977 | 0.917 |
| OmniGuard [22] | 0.853 | 0.964 | 0.810 | 0.867 | 0.973 | 0.824 | 0.868 | 0.966 | 0.830 | 0.858 | 0.965 | 0.815 | 0.864 | 0.969 | 0.823 |
| WAM [23] | 0.924 | 0.977 | 0.868 | 0.918 | 0.976 | 0.862 | 0.921 | 0.976 | 0.865 | 0.917 | 0.977 | 0.860 | 0.922 | 0.967 | 0.864 |
| Ours | **0.980** | **0.993** | **0.962** | **0.981** | **0.991** | **0.961** | **0.980** | **0.992** | **0.960** | **0.981** | **0.993** | **0.963** | **0.979** | **0.993** | **0.961** |

Table 3: Robustness comparison on the AIGC tampering dataset under different image degradation conditions (Bit Acc / F1 Score).

| Method | Clean | Gaussian Noise | | | JPEG Compression | | | Poisson Noise |
|---|---|---|---|---|---|---|---|---|
| | | σ=1 | σ=3 | σ=5 | Q=70 | Q=80 | Q=90 | |
| Stable Signature [10] | 99.10 / - | 98.51 / - | 98.23 / - | 97.90 / - | 96.12 / - | 96.73 / - | 97.19 / - | 97.47 / - |
| WOUAF [11] | 99.14 / - | 98.37 / - | 98.35 / - | 98.41 / - | 95.49 / - | 96.78 / - | 96.99 / - | 97.67 / - |
| MVSS-Net [15] | - / 0.862 | - / 0.849 | - / 0.878 | - / 0.824 | - / 0.627 | - / 0.702 | - / 0.773 | - / 0.844 |
| HDF-Net [19] | - / 0.595 | - / 0.588 | - / 0.574 | - / 0.538 | - / 0.409 | - / 0.495 | - / 0.516 | - / 0.568 |
| EditGuard [21] | 99.78 / 0.938 | 98.91 / 0.897 | 98.13 / 0.881 | 98.11 / 0.866 | 96.77 / 0.577 | 97.12 / 0.776 | 97.13 / 0.800 | **99.75** / 0.887 |
| OmniGuard [22] | 98.11 / 0.863 | 97.25 / 0.821 | 96.78 / 0.799 | 96.54 / 0.780 | 94.32 / 0.512 | 95.67 / 0.634 | 96.12 / 0.689 | 97.45 / 0.835 |
| WAM [23] | 98.17 / 0.920 | 97.56 / 0.881 | 97.12 / 0.860 | 96.89 / 0.845 | 95.78 / 0.589 | 96.45 / 0.702 | 96.78 / 0.755 | 98.12 / 0.903 |
| Ours | **99.98** / **0.972** | **99.90** / **0.940** | **99.80** / **0.939** | **99.69** / **0.928** | **99.73** / **0.908** | **99.88** / **0.927** | **99.97** / **0.951** | 99.65 / **0.947** |

in image-based positional watermarks, StableGuard adopts a more precise method for tampering localization.

We also compare the tampering localization maps generated by different methods, as shown in Fig. 4. As illustrated, StableGuard consistently achieves superior tampering detection accuracy compared to other methods. This is largely attributed to the self-supervised tampering detection framework we developed, which incorporates holistic watermarks through our MPW-VAE. This framework allows the forensic model to effectively differentiate between watermarked and non-watermarked regions. Moreover, the MoE-GFN we designed plays a pivotal role in extracting and fusing diverse forensic features, further enhancing the accuracy of tampering localization.

### 4.4 Robustness Analysis

To assess robustness against image degradations, we evaluate StableGuard on the AIGC tampering dataset under Gaussian noise, JPEG compression, and Poisson noise, following the protocol of prior works [19, 21, 22]. As shown in Tab. 3, StableGuard consistently outperforms competing methods, maintaining strong watermark extraction and tampering localization under all conditions. This robustness stems from our holistic watermark embedding strategy and the MoE-guided forensic network, which distributes watermark signals throughout the image and adaptively integrates diverse forensic cues. These results highlight StableGuard's strong potential for practical deployment.

### 4.5 Ablation Study

We conduct a comprehensive ablation study to evaluate the contributions of our core components: MPW-VAE, the MoE-GFN architecture, and the joint optimization strategy.

**Effectiveness of MPW-VAE.** To assess MPW-VAE's impact on tampering localization, we replace it with WOUAF [11], a state-of-the-art post-hoc watermarking method. Using WOUAF, we generate watermarked images (via Eq. (3)) and train MoE-GFN for tampering localization. As shown in Tab. 4 (w/o MPW-VAE), performance degrades substantially. This gap stems from two key advantages of MPW-VAE: 1) its diffusion-native, residual-based watermarking enables high-fidelity paired image generation, which significantly benefits forensic training; and 2) joint optimization with the forensic network creates a synergistic effect, enhancing both watermark embedding and tampering detection.

**Effectiveness of MoE-GFN Components.** We evaluate various architectural variants of MoE-GFN. Removing the entire MoFE block (w/o MoFE) and reverting to a plain UNet leads to subpar performance due to the lack of expert-driven feature extraction. Excluding individual experts—Watermark Extraction (w/o WEE), Tampering Localization (w/o TLE), or Boundary Enhancement (w/o BEE)—each results in notable performance drops, confirming their complementary roles

Table 4: Quantitative ablation study on the AIGC tampering dataset, where the † represents the structure used in this paper.

| Method | F1↑ | AUC↑ | IoU↑ | Bit Acc↑ | Param↓ | Flops↓ |
|---|---|---|---|---|---|---|
| w/o MPW-VAE | 0.811 | 0.796 | 0.774 | 99.13 | 52.02M | 78.51G |
| w/o MoFE | 0.931 | 0.920 | 0.905 | 95.12 | 38.11M | 45.72G |
| w/o WEE | 0.969 | 0.958 | 0.945 | 98.69 | 48.51M | 70.21G |
| w/o TLE | 0.952 | 0.940 | 0.930 | 98.90 | 48.51M | 70.21G |
| w/o BEE | 0.962 | 0.950 | 0.940 | 99.11 | 45.23M | 62.71G |
| w/o DSR | 0.966 | 0.955 | 0.948 | 98.97 | 51.26M | 75.74G |
| w/o JOS | 0.921 | 0.919 | 0.908 | 99.14 | 52.02M | 78.51G |
| Enc | 0.974 | 0.970 | 0.960 | 99.79 | 125.51M | 96.15G |
| Enc & Dec | 0.982 | 0.988 | 0.976 | 99.96 | 139.41M | 104.93G |
| Dec † | 0.980 | 0.992 | 0.961 | 99.98 | 52.02M | 78.51G |

in capturing global watermark signals, local manipulations, and boundary refinements, respectively. Replacing the Dynamic Soft Router(w/o DSR) with simple feature summation further degrades performance, highlighting its importance in adaptively fusing expert outputs based on input-specific forensic cues. While the removal of single experts (rows 3-5 in Tab. 4) leads to only modest declines in F1, AUC, IoU, and Bit Accuracy, this stems from the model operating near a performance ceiling, where marginal contributions appear limited in isolation. Crucially, the sharp performance drop when removing the entire MoFE block (row 2) highlights that these experts are complementary rather than redundant: their integration is essential for robustness and accuracy.

We also explore the placement of the MoFE block within the UNet. Placing it in the encoder (Enc) improves localization but increases computational cost and underperforms compared to the decoder placement. Using MoFE in both encoder and decoder (Enc & Dec) yields marginal gains but at the expense of significantly higher complexity. Thus, placing the MoFE block in the decoder (Dec) strikes the best balance between performance and efficiency.

**Effectiveness of Joint Optimization Strategy.** To isolate the benefit of joint self-supervised optimization, we compare StableGuard with a variant where watermark embedding/extraction and tampering localization are trained separately. Results in Tab. 4 (w/o JOS) show that decoupled training leads to weaker localization. In contrast, our unified optimization promotes stronger interaction between the embedding and forensic tasks, yielding consistent watermarks that enhance localization precision.

## 5   Conclusion

In this paper, we introduce StableGuard, a unified framework for copyright protection and tampering localization in the context of LDMs. By injecting a global watermark during the diffusion process and leveraging its intrinsic perturbation patterns, StableGuard achieves reliable and robust watermark extraction and tampering localization simultaneously. At the core of our method is the Multiplexing Watermark VAE (MPW-VAE), which generates visually consistent paired watermarked and watermark-free images to facilitate self-supervised training of the Mixture-of-Experts Guided Forensic Network (MoE-GFN). This network is composed of specialized experts that handle holistic watermark analysis, subtle artifact recognition, and boundary transition detection, with the Mixture-of-Forensic-Experts (MoFE) block ensuring precise and efficient forensic analysis. Extensive experimental evaluations validate StableGuard's superior performance, demonstrating its effectiveness and adaptability across a wide range of image scenarios, and highlighting its potential to advance both watermarking and tamper localization in diffusion models.

**Acknowledgement.** This research is supported by the "Leading Talent" under Guangdong Special Support Program (2024TX08X048), China National Key R&D Program (2024YFB4709200), Key-Area Research and Development Program of Guangzhou City (No.2023B01J0022), Guangdong Provincial Natural Science Foundation for Outstanding Youth Team Project (No. 2024B1515040010), NSFC Key Project (No. U23A20391), Guangdong Basic and Applied Basic Research Foundation (No. 2023B1515120058), Guangdong Natural Science Funds for Distinguished Young Scholars (Grant 2023B1515020097), the GuangDong Basic and Applied Basic Research Foundation (2025A1515010124), the National Research Foundation Singapore under the AI Singapore Programme (AISG Award No: AISG4-TC-2025-018-SGKR), and the Lee Kong Chian Fellowships.

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

# Appendix

## A  Problem Definition

Our goal is to imprint an imperceptible holistic watermark $W$ during the generation process of a latent diffusion model for an image $X$, and comprehensively leverage the intrinsic patterns of this watermark to achieve reliable and robust copyright protection and tampering localization within a unified framework. Let the LDMs follow a generative process parameterized by $\mathcal{G}_\phi$. The watermark embedding task can then be formulated as:

$$X = \mathcal{G}_\phi(z, W), \tag{14}$$

where $\mathcal{G}_\phi$ represents the LDM-based image generation function, and $z$ is the latent input. The embedded watermark $W$ remains imperceptible yet robust against various tampering manipulations $T$, such as copy-paste or splicing.

To detect tampering and recover the watermark, we employ a forensic network $\mathcal{F}_\psi$, which takes the potentially manipulated image $T(X)$ as input and simultaneously produces the extracted watermark $\hat{W}$ and a tampering localization map $\hat{M}$:

$$\hat{W}, \hat{M} = \mathcal{F}_\psi(T(X)). \tag{15}$$

Here, $\hat{W}$ is expected to be reliably recovered with its integrity preserved, while $\hat{M}$ precisely identifies the manipulated regions, ensuring a robust and effective framework for both watermark extraction and tampering detection.

## B  Comparison of Existing Methods

In this section, we provide a concise overview of the key comparison methods, summarized in Tab. 5. As illustrated, this work focuses on two primary tasks: tampering localization and copyright protection for content generated by diffusion models. The compared copyright protection approaches encompass both traditional post-hoc image watermarking techniques and more recent diffusion-native watermarking methods that integrate directly with the generative process. Tampering detection methods are categorized into passive localization and proactive protection strategies. Our proposed approach offers a unified, diffusion-native solution for both tampering detection and copyright protection, achieving state-of-the-art performance across both tasks.

Table 5: Comparative analysis of existing methods for copyright protection and tampering localization.

| Method | Reference | Copyright Protection | Tampering Localization | Joint Optimization | Watermarking Type |
|---|---|---|---|---|---|
| HiDDeN [7] | ECCV'18 | ✓ | | | Post-hoc |
| SepMark [8] | MM'23 | ✓ | | | Post-hoc |
| Stable Signature [10] | ICCV'23 | ✓ | | | Diffusion-native |
| WOUAF [11] | CVPR'24 | ✓ | | | Diffusion-native |
| WaDiff [12] | ECCV'24 | ✓ | | | Diffusion-native |
| MVSS-Net [15] | TPAMI'22 | | ✓ | | - |
| IML-ViT [16] | arXiv'23 | | ✓ | | - |
| PSCC-Net [17] | TCSVT'22 | | ✓ | | - |
| ObjectFormer [18] | CVPR'22 | | ✓ | | - |
| HDF-Net [19] | TPAMI'24 | | ✓ | | - |
| EditGuard [21] | CVPR'24 | ✓ | ✓ | | Post-hoc |
| OmniGuard [22] | CVPR'25 | ✓ | ✓ | | Post-hoc |
| WAM [23] | ICLR'25 | ✓ | ✓ | ✓ | Post-hoc |
| StableGuard (Ours) | NeurIPS'25 | ✓ | ✓ | ✓ | Diffusion-native |

## C  In-depth Analysis of Mixture-of-Experts Guided Forensic Network

In this paper, we introduce a Mixture-of-Experts Guided Forensic Network (MoE-GFN), which trains specialized expert modules to address distinct forensic sub-tasks, thereby enhancing the overall forensic performance. At the core of MoE-GFN is the Mixture-of-Forensic-Experts (MoFE) block,

comprising three domain-specific experts and a dynamic soft router. As illustrated in Fig. 5, we visualize the feature activations of each expert alongside the weight distributions produced by the dynamic router. The watermark extraction expert exhibits strong activation in untampered regions rich in watermark signals ($x_{\mathrm{wm}}$), effectively recovering the embedded global watermark and demonstrating robustness to local tampering. The tampering localization expert focuses on corrupted regions ($x_{\mathrm{tamp}}$), identifying missing watermark areas and enabling precise localization of tampering. The boundary enhancement expert targets the edges of tampered regions ($x_{\mathrm{bound}}$), extracting fine-grained features to refine boundary detection. Unlike static fusion strategies, the dynamic router assigns adaptive weights to each expert's features based on input characteristics ($R_{\mathrm{wm}}, R_{\mathrm{tamp}}, R_{\mathrm{bound}}$), facilitating a more effective integration of forensic cues and significantly improving both watermark extraction and tampering localization accuracy ($x_{\mathrm{unified}}$). In the Ablation Study section in the main text, we also analyzed in detail the role of each expert and dynamic fusion strategy of MoFE and the final impact on the quantization results.

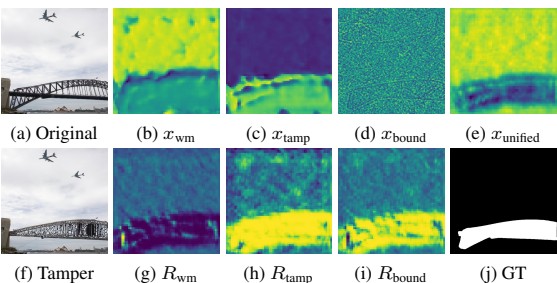

(a) Original    (b) $x_{\mathrm{wm}}$    (c) $x_{\mathrm{tamp}}$    (d) $x_{\mathrm{bound}}$    (e) $x_{\mathrm{unified}}$

(f) Tamper    (g) $R_{\mathrm{wm}}$    (h) $R_{\mathrm{tamp}}$    (i) $R_{\mathrm{bound}}$    (j) GT

Figure 5: The visualizations of the distinct features extracted by each expert and their corresponding soft weights. The Watermark Extraction Expert captures global watermark patterns, the Tampering Localization Expert focuses on fine-grained artifacts, and the Boundary Enhancement Expert sharpens tampering boundaries. These complementary features are adaptively fused via dynamic weighting, enabling comprehensive and precise forensic analysis.

## D   Training Process of StableGuard

We present the detailed training procedure of StableGuard in Algorithm 1. The entire process is fully self-supervised and trained in an end-to-end manner, without requiring any manually annotated data. Both the embedding of diffusion-native watermarks and the training of the joint model for watermark and tampering forensics are accomplished within a unified framework.

---

**Algorithm 1** Training Process of StableGuard.

---

**Input:** Input image $X$, iteration steps $steps$, LDM VAE encoder $\mathcal{E}_\mu$, LDM VAE decoder $\mathcal{D}_\nu$.
**Output:** MPW-VAE decoder $\mathcal{D}_\phi$, MoE-GFN $\mathcal{F}_\psi$.

1: **for** $i = 1$ **to** $steps$ **do**
2:     Inverse latent $z = \mathcal{E}_\mu(X)$.
3:     Decode watermark-free image $\hat{X} = \mathcal{D}_\nu(z)$.
4:     Decode watermarked image $Y = \mathcal{D}_\phi(z)$.
5:     Generate random tamper mask $M$.
6:     Generate tamper image $\hat{Y}$ by Eq. (3).
7:     Predict watermark and tamper area $\hat{W}, \hat{M} = \mathcal{F}_\psi(\hat{Y})$
8:     Comptue loss by Eq. (13).
9:     Update $\mathcal{D}_\phi$ and $\mathcal{F}_\psi$.
10: **end for**

---

# E More Implementation Details and Experiments

## E.1 More Implementation Details

In our experiments, we utilized two datasets: one consisting of real images from COCO [56], and the other comprising generated images from a Text-to-Image dataset. For the real image dataset, we first reconstructed each image using our MPW-VAE and then embedded a holistic watermark into the reconstructed image. For the generated dataset, we began by obtaining the denoised latent representation through stable diffusion, which was then decoded into the image space using the MPW-VAE decoder to produce the watermarked image.

**Text-to-Image (T2I) Dataset**. The T2I dataset is designed to evaluate the watermarking effectiveness of StableGuard in the LDM framework. To construct this dataset, we first generate the latent embedding of each image using the original pre-trained Stable Diffusion [1] denoising UNet, condition on its corresponding textual prompt. We then decode this embedding through our MPW-VAE, producing a set of watermarked images. Specifically, we leverage ChatGPT[2] to generate 1,000 textual prompts, with each prompt yielding 10 images, resulting in a dataset of 10,000 images.

**AIGC Tampering Dataset**.[3] The AIGC tampering dataset is collected to benchmark the performance of various tampering localization techniques against manipulations introduced by state-of-the-art image editing methods [1, 2, 4, 57, 58]. This large-scale dataset consists of 35,000 images, comprising 25,000 from COCO and 10,000 from our T2I dataset. We utilize the Segment Anything Model (SAM) [49] to generate semantic masks by segmenting images, after which we apply Stable Diffusion Inpainting [1], Stable Diffusion XL [2], Kandinsky Inpainting [57], ControlNet [4], and LAMA [58] to edit the segmented regions. During the tampering process, we applied modifications to selected regions and composited the altered content onto the watermarked image using a corresponding binary mask following by [21, 22].

**Metrics.** We leverage several widely-used metrics to collaboratively evaluate the pixel-level fidelity and perceptual quality of the generated images, including Peak Signal-to-Noise Ratio (PSNR), Structural Similarity Index Measure (SSIM) [60], Learned Perceptual Image Patch Similarity (LPIPS) [54], and Fréchet Inception Distance (FID) [61]. To quantify the performance of watermark extraction, we adopt the Bit Accuracy metric to calculate the proportion of correctly extracted watermark bits over the total embedded bits. For tampering detection, we used the F1 Score, the Area Under Curve (AUC), and IoU metrics to assess the accuracy and completeness of the detected area. The F1 Score is the harmonic mean of detection precision and recall, and AUC represents the area under the ROC Curve. IoU is calculated by the intersection over the union between the detected tampered areas and the ground-truth regions.

**Forensic Decoder.** The forensic decoder consists of two parallel decoding heads: 1) The mask prediction head is responsible for localizing tampered regions and is implemented as a lightweight two-layer convolutional network. 2) The watermark prediction head is used for recovering the embedded watermark and comprises a two-layer convolutional network followed by a fully connected layer. This modular design allows the network to jointly optimize for both localization and watermark reconstruction.

## E.2 Tampering Localization Results on Traditional Tampering Dataset

Our proposed StableGuard framework can be naturally extended to conventional natural images through reconstruction using a variational autoencoder, demonstrating strong generalization across a wide range of image domains. To assess its effectiveness beyond synthetic settings, we conducted comprehensive evaluations on five widely used tampering datasets: CASIA [62], NIST [63], Columbia [64], and Coverage [65]. The corresponding quantitative results and visual comparisons are reported in Tab. 6 and Fig. 6.

Although existing passive tampering localization methods are trained on meticulously constructed paired datasets, they consistently fall short when compared to our StableGuard and its predecessor EditGuard, OmniGuard and WAM. This performance gap reveals the inherent limitations of passive, post-hoc approaches, which often lack robustness when faced with novel manipulations or diverse

---

[2]https://chat.openai.com/
[3]The full dataset is available at https://github.com/Harxis/StableGuard.

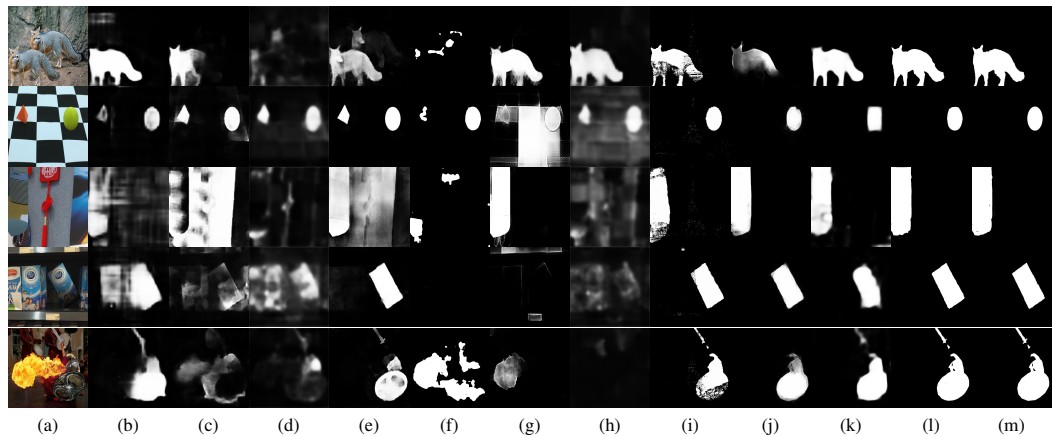

|  | (a) | (b) | (c) | (d) | (e) | (f) | (g) | (h) | (i) | (j) | (k) | (l) | (m) |

Figure 6: The visualization comparison of tampering localization results on the traditional tampering dataset. (a) is the tampered image, (b)-(m) are the tampering localization results of MVSS-Net [15], PSCC-Net [17], ObjectFormer [18], TruFor [66], NCL-IML [67], IML-ViT [16], HDF-Net [19], EditGuard [21], OmniGuard [22], WAM [23], our StableGuard, and the ground truth, respectively.

Table 6: Localization precision comparison with other competitive methods on five common datasets [62–65, 68].

| Method | CAISA1.0 [62] | | | NIST16 [63] | | | Columbia [64] | | | Coverage [65] | | | IMD20 [68] | | |
|---|---|---|---|---|---|---|---|---|---|---|---|---|---|---|---|
| | F1↑ | AUC↑ | IoU↑ | F1↑ | AUC↑ | IoU↑ | F1↑ | AUC↑ | IoU↑ | F1↑ | AUC↑ | IoU↑ | F1↑ | AUC↑ | IoU↑ |
| Mantra-Net [69] | 0.215 | 0.661 | 0.120 | 0.428 | 0.828 | 0.272 | 0.351 | 0.693 | 0.213 | 0.262 | 0.712 | 0.159 | 0.271 | 0.731 | 0.157 |
| SPAN [70] | 0.252 | 0.676 | 0.144 | 0.205 | 0.600 | 0.114 | 0.465 | 0.741 | 0.303 | 0.258 | 0.659 | 0.148 | 0.217 | 0.701 | 0.134 |
| CAT-Net [71] | 0.258 | 0.717 | 0.184 | 0.695 | 0.948 | 0.573 | 0.418 | 0.460 | 0.264 | 0.292 | 0.739 | 0.194 | 0.184 | 0.452 | 0.102 |
| MVSS-Net [15] | 0.263 | 0.638 | 0.207 | 0.833 | 0.982 | 0.756 | 0.424 | 0.764 | 0.276 | 0.229 | 0.816 | 0.152 | 0.318 | 0.703 | 0.234 |
| PSCC-Net [17] | 0.212 | 0.611 | 0.136 | 0.479 | 0.846 | 0.315 | 0.532 | 0.758 | 0.363 | 0.297 | 0.723 | 0.174 | 0.229 | 0.707 | 0.145 |
| ObjectFormer [18] | 0.208 | 0.739 | 0.135 | 0.720 | 0.958 | 0.642 | 0.529 | 0.765 | 0.374 | 0.277 | 0.756 | 0.165 | 0.251 | 0.802 | 0.166 |
| TruFor [66] | 0.445 | 0.844 | 0.286 | 0.360 | 0.806 | 0.220 | 0.566 | 0.869 | 0.395 | 0.119 | 0.837 | 0.063 | 0.217 | 0.810 | 0.122 |
| NCL-IML [67] | 0.169 | 0.766 | 0.123 | 0.885 | 0.981 | 0.834 | 0.420 | 0.712 | 0.253 | 0.262 | 0.824 | 0.172 | 0.339 | 0.827 | 0.245 |
| HDF-Net [19] | 0.395 | 0.747 | 0.303 | 0.895 | **0.989** | 0.840 | 0.600 | 0.820 | 0.513 | 0.345 | 0.811 | 0.233 | 0.355 | 0.814 | 0.257 |
| EditGuard [21] | **0.914** | 0.951 | 0.865 | 0.921 | 0.957 | 0.881 | 0.953 | 0.954 | 0.943 | 0.945 | 0.964 | 0.928 | 0.906 | 0.926 | 0.851 |
| OmniGuard [22] | 0.878 | 0.942 | 0.812 | 0.910 | 0.953 | 0.857 | 0.941 | 0.950 | 0.918 | 0.920 | 0.958 | 0.895 | 0.870 | 0.912 | 0.801 |
| WAM [23] | 0.872 | 0.930 | 0.803 | 0.908 | 0.950 | 0.854 | 0.938 | 0.947 | 0.912 | 0.918 | 0.955 | 0.891 | 0.865 | 0.908 | 0.792 |
| Ours | 0.933 | **0.970** | **0.900** | **0.956** | 0.983 | **0.926** | **0.995** | **0.997** | **0.990** | **0.986** | **0.994** | **0.973** | **0.949** | **0.975** | **0.919** |

Table 7: Statistics of runtime latency and VRAM on the AIGC tampering dataset.

| Metric | Vanilla VAE | MPW-VAE | MoE-GFN |
|---|---|---|---|
| Latency | 12.69 ms | 14.99 ms | 91.88 ms |
| VRAM | 1681.93 MB | 2033.97 MB | 428.32 MB |

image distributions. In contrast, our findings underscore the advantages of proactive forensic methods, particularly those grounded in watermarking strategies that are embedded during the generative process. These methods exhibit greater resilience and accuracy in both tampering detection and localization.

In addition, StableGuard demonstrates superior performance over other methods across a variety of practical scenarios. It achieves this improvement while eliminating the need for complex and often restrictive post-processing procedures. This enhancement is largely attributed to the Mixture of Experts Guided Forensic Network, which is designed to extract and integrate diverse forensic cues. By employing specialized expert modules and adaptive feature routing, the network enables StableGuard to jointly perform watermark recovery and tampering localization in a coherent and dependable manner.

Table 8: Comparison of watermark extraction accuracy (↑) under different tampering rates on the AIGC tampering dataset.

| Method | 10% | 30% | 50% | 70% | 90% |
|---|---|---|---|---|---|
| EditGuard [21] | 99.78 | 99.60 | 97.66 | 90.95 | 69.13 |
| OmniGuard [22] | 98.11 | 98.02 | 96.84 | 91.33 | 83.90 |
| WAM [23] | 98.17 | 97.11 | 94.89 | 93.74 | 88.53 |
| Ours | **99.98** | **99.98** | **99.96** | **99.27** | **89.58** |

Table 9: Comparison of false positive rate (FPR ↓) on the AIGC tampering dataset.

| Type | EditGuard [21] | OmniGuard [22] | WAM [23] | Ours |
|---|---|---|---|---|
| Partially tampered | 0.0422 | 0.0191 | 0.0045 | **0.0023** |
| Fully untampered | 0.0028 | 0.0024 | 0.0019 | **0.0016** |

## E.3 Time Complexity Analysis

We conducted additional experiments to evaluate the inference latency and memory footprint of our framework. The detailed results are summarized in Tab. 7. Specifically, we measured the runtime and memory usage of our MPW-VAE in comparison to the baseline Stable Diffusion VAE (at a resolution of $512 \times 512$), as well as the runtime of the MoE-GFN forensic network under identical settings. The results show that MPW-VAE introduces only negligible computational overhead relative to the original diffusion VAE. Moreover, the MoE-GFN model achieves competitive runtime efficiency, with both inference latency and memory consumption well within the bounds required for practical deployment.

## E.4 Forensic Accuracy under Varying Tampering Ratios

We further analyze watermark extraction accuracy under different levels of tampering, ranging from 10% to 90%. The results in Tab. 8 demonstrate that our method consistently outperforms existing approaches across all tampering ratios. This robustness stems from three main factors: 1) The MPW-VAE effectively encodes holistic watermark features at multiple scales during decoding, enhancing the resilience of watermark embedding. 2) The global self-attention mechanism in our watermark extraction expert captures long-range dependencies and contextual cues that remain intact even when large regions are tampered. 3) The use of randomly generated masks with varying coverage during training, including large-area tampering, encourages the model to generalize across diverse manipulation ratios and ensures reliable watermark extraction even under severe distortions.

## E.5 True Positive Rate (TPR) under Tampering Scenarios

Our framework is designed as a proactive watermarking system [21–23], where the watermark plays a central role in simultaneously enabling copyright protection and tampering localization. Unlike passive detection methods, we assume that images are watermarked at creation time; thus, clean images without watermarks fall outside our system's operational scope and can be conservatively regarded as "fully tampered." To assess practical reliability, we measure false positives in two scenarios: (1) untampered regions within partially tampered watermarked images, and (2) fully untampered, watermarked images. As shown in Tab. 9, our method achieves the lowest false positive rate (measured as the ratio of false positive pixels to the total number of untampered pixels) in both cases, demonstrating its robustness and reliability.

## E.6 Performance across Different Tampering Types

We also conducted a per-type analysis of tampering detection accuracy, as reported in Tab. 10. Our approach consistently delivers strong performance across diverse manipulation types, underscoring its robustness and generalizability.

Table 10: Tampering performance of our method under different tampering types on the AIGC tampering dataset.

| Type | SD-Inpainting [1] | | | SD-XL [2] | | | Kandinsky [57] | | | ControlNet [4] | | | LAMA [58] | | |
|---|---|---|---|---|---|---|---|---|---|---|---|---|---|---|---|
| | F1↑ | AUC↑ | IoU↑ | F1↑ | AUC↑ | IoU↑ | F1↑ | AUC↑ | IoU↑ | F1↑ | AUC↑ | IoU↑ | F1↑ | AUC↑ | IoU↑ |
| Splicing | 0.975 | 0.991 | 0.963 | 0.984 | 0.994 | 0.965 | 0.985 | 0.993 | 0.964 | 0.984 | 0.994 | 0.963 | 0.982 | 0.995 | 0.965 |
| Copy-and-paste | 0.978 | 0.990 | 0.958 | 0.976 | 0.986 | 0.959 | 0.980 | 0.989 | 0.961 | 0.978 | 0.987 | 0.957 | 0.976 | 0.991 | 0.960 |
| Removal | 0.980 | 0.994 | 0.965 | 0.982 | 0.989 | 0.966 | 0.978 | 0.990 | 0.962 | 0.981 | 0.993 | 0.964 | 0.984 | 0.988 | 0.968 |
| Inpainting | 0.987 | 0.997 | 0.962 | 0.982 | 0.995 | 0.954 | 0.977 | 0.996 | 0.953 | 0.981 | 0.998 | 0.968 | 0.974 | 0.998 | 0.951 |

Table 11: Performance under various degradations on AIGC tampering dataset.

| Type | EditGuard [21] | | OmniGuard [22] | | WAM [23] | | Ours | |
|---|---|---|---|---|---|---|---|---|
| | Bit Acc.↑ | F1↑ | Bit Acc.↑ | F1↑ | Bit Acc.↑ | F1↑ | Bit Acc.↑ | F1↑ |
| Clean | 99.78 | 0.938 | 98.11 | 0.863 | 98.17 | 0.920 | **99.98** | **0.972** |
| JPEG (Q=30) | 62.30 | 0.230 | 63.34 | 0.311 | 80.96 | 0.532 | **98.87** | **0.866** |
| WebP (Q=50) | 52.88 | 0.233 | 56.16 | 0.307 | 84.87 | 0.538 | **98.95** | **0.703** |
| HEIC (Q=50) | 66.53 | 0.335 | 59.41 | 0.323 | 87.46 | 0.402 | **98.36** | **0.748** |
| Resale (0.25) | 41.77 | 0.256 | 54.95 | 0.343 | 95.98 | 0.821 | **97.55** | **0.858** |
| Brightness (0.8-1.2) | 91.35 | 0.504 | 89.55 | 0.741 | 94.71 | 0.816 | **98.94** | **0.860** |
| Contrast (0.8-1.2) | 90.84 | 0.788 | 91.45 | 0.836 | 95.00 | 0.824 | **97.93** | **0.836** |
| Saturation (0.8-1.2) | 92.93 | 0.817 | 93.44 | 0.856 | 95.69 | 0.814 | **97.98** | **0.911** |

## E.7 Robustness under Real-World Image Degradations

To evaluate real-world applicability, we conducted additional experiments with diverse image degradation operations, including: (1) aggressive compression, (2) severe resolution downsampling, and (3) color transformations. The parameter ranges for each transformation are summarized in Tab. 11. As shown in the results, our framework maintains high watermark extraction accuracy and reasonable tampering localization performance. This observation aligns with our design intuition: watermark extraction leverages global cues that are relatively robust to distortions, whereas tampering localization relies on local consistency, making it more sensitive to compression and resampling artifacts. Importantly, the decline in localization accuracy remains moderate and within acceptable limits, highlighting the resilience of our method in real-world scenarios.

## E.8 More Visual Results

Fig. 7 and Fig. 8 present additional visual results of tampering localization on the AIGC tampering dataset of COCO [56] and T2I Dataset, respectively. Please zoom in for the best view.

## F  Disscussion and Limitations

StableGuard is specifically designed for image latent diffusion models. However, since watermarking is applied solely through the variational autoencoder decoder within the latent diffusion framework, our approach can be naturally extended to video latent diffusion models. Given the rapid progress in this area, adapting our method to the video domain represents an important direction for future research.

While our approach maintains strong performance under certain levels of image degradation, it is not entirely immune to the decline in forensic accuracy caused by such distortions. Addressing the sensitivity of forensic performance to image degradation remains a current limitation of our method and a key focus for future investigation.

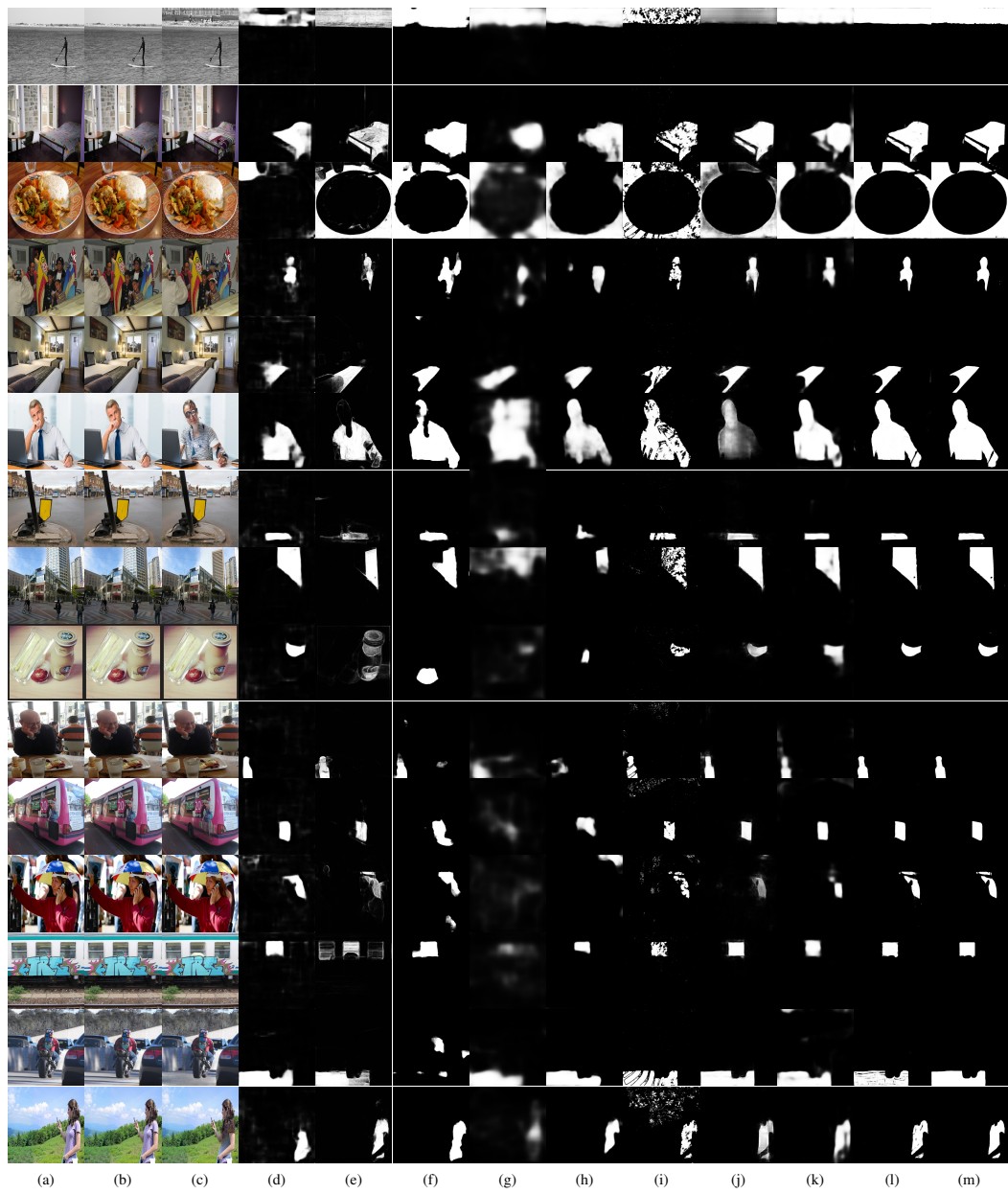

Figure 7: The visualization of tampering localization results on the AIGC tampering dataset of COCO part. (a) is the original image, (b) is the watermarked image by StableGuard, (c) is the tampered image, (d)-(m) are the tampering localization results of MVSS-Net [15], IML-ViT [16], PSCC-Net [17], ObjectFormer [18], HDF-Net [19], EditGuard [21], OmniGuard [22], WAM [23], our StableGuard, and the ground truth, respectively.

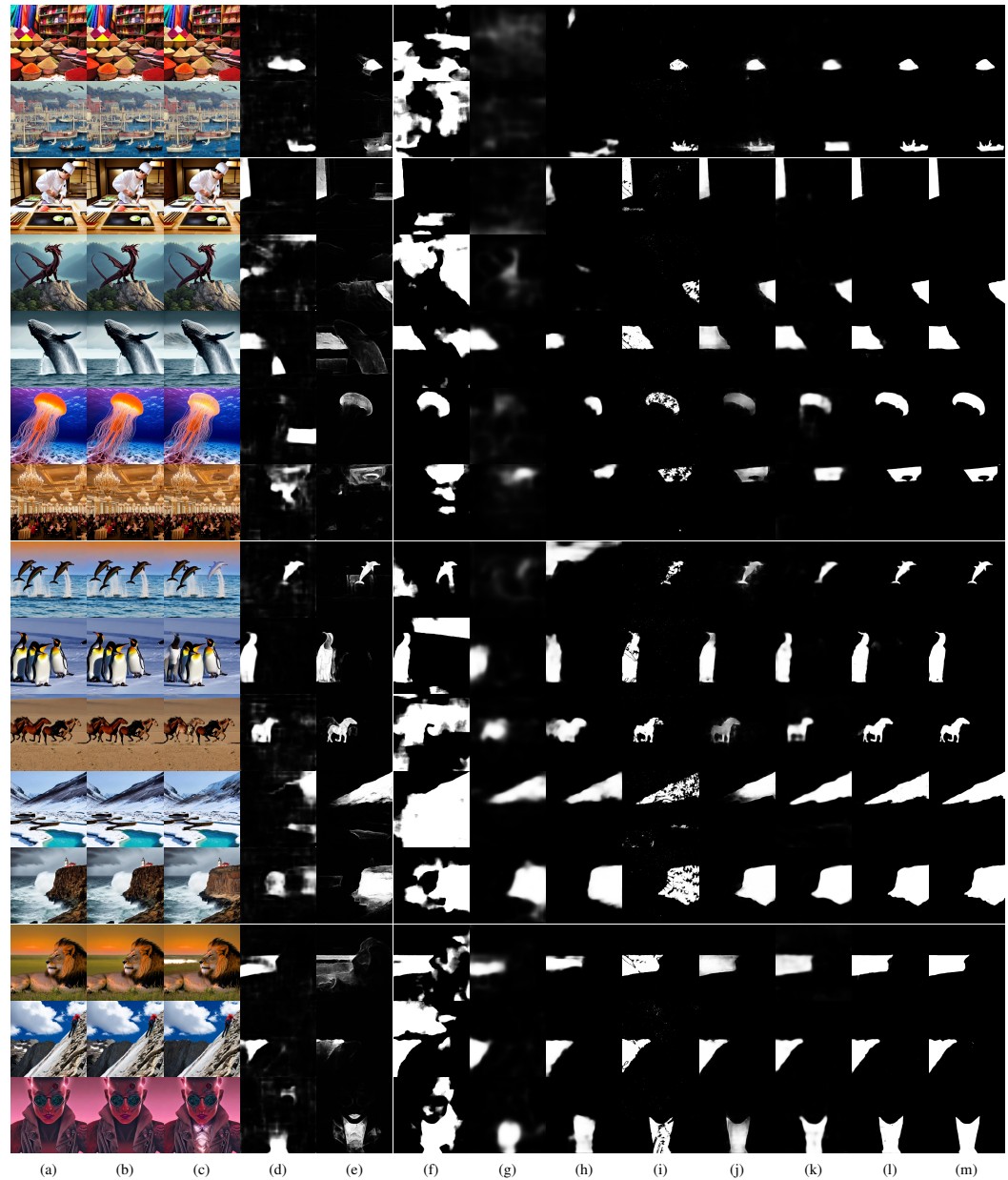

Figure 8: The visualization of tampering localization results on the AIGC tampering dataset of T2I dataset part. (a) is the original image, (b) is the watermarked image by StableGuard, (c) is the tampered image, (d)-(m) are the tampering localization results of MVSS-Net [15], IML-ViT [16], PSCC-Net [17], ObjectFormer [18], HDF-Net [19], EditGuard [21], OmniGuard [22], WAM [23], our StableGuard, and the ground truth, respectively.

