# OpenReview forum: "StableGuard: Towards Unified Copyright Protection and Tamper Localization in Latent Diffusion Models"
_NeurIPS.cc/2025/Conference — NeurIPS 2025 poster_

### Official Review · Reviewer_4kgq · 2025-06-25

**Clarity:** 3
**Significance:** 2
**Originality:** 2
**Rating:** 4
**Confidence:** 5

**Summary:**

This paper presents StableGuard, a unified framework for copyright protection and tampering localization based on a global watermark embedded during the diffusion process. The MPW-VAE module ensures that the embedded watermark maintains high visual fidelity in the watermarked images. In addition, the MoE-GFN module integrates three experts: Watermark Extraction, Tampering Localization, and Boundary Detection. These modules work together to improve the accuracy of forensic analysis.

**Questions:**

1. In Equation (3), why is the fusion based on a random choice between $X$and$\hat{X}$? It seems that using either one consistently might be sufficient.

2. The paper provides detailed descriptions of each module. However, the architecture of the forensic decoder is not clearly explained. Could the authors briefly describe its structure?

[1]Sander, Tom, et al. "Watermark Anything with Localized Messages." Proceedings of the International Conference on Learning Representations. 2025.

**Ethical Concerns:**

["NO or VERY MINOR ethics concerns only"]

**Final Justification:**

The authors have addressed my concern about scenarios under different tempering rates. Besides, the comparison with WAM also shows their method’s superior performance. Therefore, I increase my score.

**Limitations:**

yes

**Quality:**

3

**Strengths And Weaknesses:**

### Strengths

1. The paper is clearly written, with detailed explanations of the proposed method and its components.

2. The work introduces a diffusion-based watermark for tampering localization by embedding watermarks during the image generation process. This provides a valuable supplement to the field.

### Weaknesses

1. The proposed method uses a single watermark to achieve both copyright protection and tampering localization, which involves a trade-off between robustness and localization sensitivity. Although the authors employ two expert modules for extracting different features, it seems that the final decision relies on $x_{unified}$​. When the tampered region becomes large, this may reduce watermark extraction accuracy. It would be helpful if the authors could report the performance of watermark extraction under different tampering ratios.

2. The robustness of the method appears to be limited or requires further evaluation. Methods like WAM[1] also support both tampering localization and watermark extraction with a single watermark, and are able to resist a wide range of transformations. A comparison or discussion would strengthen the contribution.

---

> ### Author Rebuttal · Authors · 2025-07-27
>
> We thank the reviewer for their thoughtful evaluation and constructive comments. While the points raised are valid and appreciated, they focus on peripheral implementation details that we clarify below.
>
> > **Q1**: The proposed method uses a single watermark to achieve both copyright protection and tampering localization, which involves a trade-off between robustness and localization sensitivity. Although the authors employ two expert modules for extracting different features, it seems that the final decision relies on $x_{unified}$. When the tampered region becomes large, this may reduce watermark extraction accuracy. It would be helpful if the authors could report the performance of watermark extraction under different tampering ratios.
>
>  **A1**: We appreciate the comment. We have conducted a comparative analysis of watermark extraction accuracy across different tampering ratios, ranging from 10% to 90%. The results, presented in the table below, show that our method consistently outperforms existing approaches across all tampering levels. This remarkable performance can be attributed to three key factors:
>  1) Our MPW-VAE seamlessly encodes holistic watermark features at multiple scales during the decoding process, which significantly enhances the robustness of watermark embedding.
>  2) The global self-attention mechanism in our watermark extraction expert facilitates the capture of long-range dependencies and contextual cues that remain even when large regions are tampered.
>  3) We introduce randomly generated masks with varying coverage during the training process, including large-area tampering. This encourages the model to generalize well to a wide range of manipulation ratios and improves its ability to extract watermarks even under severe distortions.
>
>  We will include these results in the revision.
>
> **Table 1**. Comparison of watermark extraction accuracy (↑) under different tampering rates on the AIGC tampering dataset.
> |     Method   |     10%       |     30%       |     50%       |     70%       |     90%       |
> |------------------|--------------|--------------|--------------|--------------|--------------|
> |     EditGuard    |     99.78    |     99.60    |     97.66    |     90.95    |     69.13    |
> |     OmniGuard    |     98.11    |     98.02    |     96.84    |     91.33    |     83.90    |
> |     WAM          |     98.17    |     97.11    |     94.89    |     93.74    |     88.53    |
> |     Ours     |     **99.98**    |     **99.98**    |     **99.96**    |     **99.27**    |     **89.58**    |
>
>
> > **Q2**: The robustness of the method appears to be limited or requires further evaluation. Methods like WAM [1] also support both tampering localization and watermark extraction with a single watermark, and are able to resist a wide range of transformations. A comparison or discussion would strengthen the contribution.
>
>  **A2**: We would first like to highlight a fundamental distinction between our approach and WAM [1]. WAM adopts a post-hoc watermarking strategy, injecting watermarks into pre-generated images, while our method is diffusion-naive and jointly optimized, embedding holistic watermark features directly during the generative process. This design allows for seamless integration of watermark signals with the image content and enables improved resilience to tampering.
>
>  In response to your suggestion and in accordance with the comments from Reviewer **#Fd7Y**, we performed a comparative evaluation against two recent representative methods, namely WAM [1] and OmniGuard [2], both of which perform tampering localization using a single post-hoc embedded watermark. As presented in the tables below, our approach consistently outperforms both methods across all evaluated metrics.
>
>  Additionally, following the suggestion from Reviewer **#DW9k**, we have included results for our method under various real-world image transformations. Please see **A4** under **#DW9k** for details.
>
>  We will include these results, as well as additional visualizations and discussion, in the revised version.
>
>
>
> **Table 2**: Quantitative comparison on watermarking performance on COCO and the T2I dataset.
> |     Method       |     B.L.    |     COCO     |              |              |             |                 |     T2I     |              |              |             |                 |
> |------------------|-------------|---------------------|--------------|--------------|-------------|-----------------|----------------|-----------|--------------|-------------|-------------|
> |                  |             |     PSNR↑            |     SSIM↑     |     LPIPS↓    |     FID↓     |     Bit Acc.↑    |     PSNR↑           |     SSIM↑     |     LPIPS↓    |     FID↓     |     Bit Acc.↑    |
> |     OmniGuard    |     100     |     37.54           |     0.950    |     0.072    |     20.1    |     98.11       |     37.32          |     0.944    |     0.070    |     20.0    |     98.09       |
> |     WAM          |     32      |     38.20           |     0.951    |     0.067    |     19.9    |     98.17       |     37.80          |     0.946    |     0.073    |     19.6    |     98.33       |
> |     Ours     |     32      |     **40.50**           |     **0.970**    |     **0.062**    |     **19.5**    |     **99.97**       |     **40.53**          |     **0.972**    |     **0.060**    |     **19.4**    |     **99.98**       |
> |     Ours     |     128     |     40.10           |     0.966    |     0.070    |     19.9    |     99.87       |     40.11          |     0.968    |     0.069    |     19.8    |     99.88       |
>
> **Table 3**: Localization precision comparison on the AIGC tampering dataset.
> |     Method       |     SD-Inp.    |              |              |     SD-XL    |              |              |     Kand.    |              |              |     Cont.    |              |              |     LAMA     |              |              |
> |-------------|:-----------------:|:-----------:|:-----------:|:-----------:|:--------------:|:--------------:|:------------------:|:--------------:|:--------------:|:----------------:|:--------------:|:--------------:|:--------------:|:--------------:|:--------------:|
> |                  |     F1↑               |     AUC↑      |     IoU↑      |     F1↑       |     AUC↑      |     IoU↑      |     F1↑           |     AUC↑      |     IoU↑      |     F1↑            |     AUC↑      |     IoU↑      |     F1↑       |     AUC↑      |     IoU↑      |
> |     OmniGuard    |     0.853            |     0.964    |     0.810    |     0.867    |     0.973    |     0.824    |     0.868        |     0.966    |     0.830    |     0.858         |     0.965    |     0.815    |     0.864    |     0.969    |     0.823    |
> |     WAM          |     0.924            |     0.977    |     0.868    |     0.918    |     0.976    |     0.862    |     0.921        |     0.976    |     0.865    |     0.917         |     0.977    |     0.860    |     0.922    |     0.967    |     0.864    |
> |   Ours       |     **0.980**            |     **0.993**    |     **0.962**    |     **0.981**    |     **0.991**    |     **0.961**    |     **0.980**        |     **0.992**    |     **0.960**    |     **0.981**         |     **0.993**    |     **0.963**    |     **0.979**    |     **0.993**    |     **0.961**    |
>
>
>
> > **Q3**: In Equation (3), why is the fusion based on a random choice between $X$ and $\hat{X}$. It seems that using either one consistently might be sufficient.
>
>  **A3**:  The random selection between $X$ and $\hat{X}$ in Equation (3) is intentionally designed to improve the generalization ability of the model across a broader spectrum of tampering types.
>
>  Specifically, choosing $X$ (a real image) allows us to simulate conventional tampering patterns typically observed in human-made manipulations, such as those created using image editing tools (e.g., Photoshop). Conversely, using $\hat{X}$, which is the output reconstructed by the vanilla LDM's VAE, mimics AI-generated forgeries that are increasingly prevalent with the rise of generative models.
>
>  By incorporating both sources in a randomized fashion during training, the model is exposed to a wider distribution of tampering artifacts, spanning both traditional and generative forgery patterns. This strategy enables the model to learn more robust representations and improves its applicability to diverse real-world scenarios.
>
>  We will incorporate a more detailed explanation of this design choice in the revision.
>
> > **Q4**: The paper provides detailed descriptions of each module. However, the architecture of the forensic decoder is not clearly explained. Could the authors briefly describe its structure?
>
>  **A4**: The forensic decoder consists of two parallel decoding heads:
>  1) The mask prediction head is responsible for localizing tampered regions and is implemented as a lightweight two-layer convolutional network.
>  2) The watermark prediction head is used for recovering the embedded watermark and comprises a two-layer convolutional network followed by a fully connected layer.
>
> This modular design allows the network to jointly optimize for both localization and watermark reconstruction. We will provide a comprehensive description of the forensic decoder in the revised version and make our code, pretrained models, and datasets publicly available to facilitate reproducibility upon acceptance.
>
>  [1] *Sander T, et al. Watermark Anything With Localized Messages, ICLR 2025.* \
>  [2] *Zhang X, et al. Omniguard: Hybrid manipulation localization via augmented versatile deep image watermarking, CVPR 2025.*
>
>  > **As all concerns have been thoroughly addressed and none pertain to the central novelty, significance, or empirical effectiveness of our approach, we respectfully believe that the borderline rejection rating may not accurately reflect the strength of the submission. We kindly ask the reviewer to reconsider their score in light of our detailed responses.**

---

> > ### Comment · Reviewer_4kgq · 2025-08-06
> >
> > Thanks for the rebuttal. The authors have addressed my concern about scenarios under different tempering rates. Besides, the comparison with WAM also shows their method’s superior performance. I have no further questions. Therefore, I have increased my score to borderline accept.

---

> > > ### Author Response · Authors · 2025-08-06
> > >
> > > We sincerely appreciate your valuable suggestions in the review, which have helped improve our work, as well as your recognition of our efforts.

---

### Official Review · Reviewer_9vPj · 2025-06-30

**Clarity:** 3
**Significance:** 3
**Originality:** 3
**Rating:** 5
**Confidence:** 4

**Summary:**

This work proposes a method for both copyright protection and image manipulation localization using Stable Diffusion models. Specifically, it introduces two novel components: (1) MPW-VAE, which generates both watermarked and watermark-free images for self-supervised learning, and (2) MoE-GFN, which is designed for watermark verification and manipulation localization.

**Questions:**

1) The paper mentions that random splicing is applied after MPW-VAE. Could the authors clarify why only splicing is considered? What would be the effect of applying other manipulation types, such as random copy-move or random removal, in this step?
2) What is the approximate inference time of the proposed model for processing a single image, including both watermark verification and manipulation localization?

**Ethical Concerns:**

["NO or VERY MINOR ethics concerns only"]

**Final Justification:**

After reviewing the rebuttal, I believe the authors have satisfactorily addressed my concerns. I consider this a solid piece of work and maintain my rating of Accept.

**Limitations:**

Yes

**Quality:**

3

**Strengths And Weaknesses:**

Strengths:
1) The paper is well-written, with clear and detailed descriptions of the proposed method.
2) The proposed approach achieves improved performance over state-of-the-art methods.
3) The experiments and ablation studies are thorough and well-designed.

Weaknesses:
1) The use of self-supervised learning has been explored in prior works [1][2], but these related methods are not discussed in the related work section.
2) Table 2 does not specify the manipulation types or the number of samples per type for the evaluated datasets.
3) The performance across different manipulation types should be reported to better understand the method's effectiveness in various scenarios.



[1] Zhai, Yuanhao, Tianyu Luan, David Doermann, and Junsong Yuan. "Towards generic image manipulation detection with weakly-supervised self-consistency learning." In Proceedings of the IEEE/CVF International Conference on Computer Vision, pp. 22390-22400. 2023.

[2] Zhang, Zhenfei, Mingyang Li, Xin Li, Ming-Ching Chang, and Jun-Wei Hsieh. "Image Manipulation Detection with Implicit Neural Representation and Limited Supervision." In European Conference on Computer Vision, pp. 255-273. Cham: Springer Nature Switzerland, 2024.

---

> ### Author Rebuttal · Authors · 2025-07-27
>
> Thank you for your insightful comments and valuable suggestions. We have revised our paper based on your feedback. Here are our responses to your comments:
>
> > **Q1**: The use of self-supervised learning has been explored in prior works [1][2], but these related methods are not discussed in the related work section.
>
>  **A1**: Thank you for suggesting the relevant works. While both [1] and [2] explore semi-supervised tampering localization, their problem settings and methodological designs differ substantially from ours.
>
>  1) Our proposed StableGuard is a unified framework that integrates both proactive watermarking and tampering localization, whereas [1] and [2] focus exclusively on passive tampering detection.
>  2) StableGuard operates in a proactive protection setting, where imperceptible watermarks are injected into protected images in advance. In contrast, [1] and [2] are post-hoc forensic methods that analyze tampering without any prior intervention or watermark signal.
>  3) Our method employs a fully self-supervised training paradigm, requiring no manual annotations. In comparison, [1] and [2] adopt semi-supervised learning and still rely on image-level supervision.
>
>  We will incorporate a more thorough discussion of these works and their distinctions from ours in the revision.
>
> > **Q2**: Table 2 does not specify the manipulation types or the number of samples per type for the evaluated datasets.
>
>  **A2**: The AIGC tampering dataset used for evaluation consists of 35,000 manipulated images, generated by five representative generative models and evenly distributed across four common tampering types: splicing, copy-and-paste, object removal, and inpainting. Each tampering category contains 8,750 samples, enabling a balanced and comprehensive evaluation across manipulation types.
>
>  We will include this clarification in the revised manuscript.
>
> > **Q3**: The performance across different manipulation types should be reported to better understand the method's effectiveness in various scenarios.
>
>  **A3**: In our original manuscript, we strictly followed the evaluation protocol established in prior works [3,4,5], where average performance across manipulation types is reported as the primary metric. This choice was made to ensure an intuitive and fair comparison with existing methods, while adhering to the space constraints of the main paper.
>
>  Nevertheless, we have conducted a per-type performance analysis, as presented in Table 1 below. The results indicate that our method consistently achieves strong performance across all manipulation types, underscoring its robustness and generalizability in diverse tampering scenarios.
>
>  Moreover, *Appendix Section F* of our original manuscript also presents additional experiments on real-world tampering datasets covering a variety of manipulation scenarios, which further support the generalizability and effectiveness of our method in practical applications.
>
>  We will include these results and the corresponding discussions in the revision.
>
>
> Table 1. Tampering performance of our method under different tampering types on the AIGC tampering dataset.
> |     Type       |     SD-Inp.    |              |              |     SD-XL    |              |              |     Kand.    |              |              |     Cont.    |              |              |     LAMA     |              |              |
> |------------------|:----------------------:|:--------------:|:--------------:|:--------------:|:--------------:|:--------------:|:------------------:|:--------------:|:--------------:|:-------------------:|:--------------:|:--------------:|:--------------:|:--------------:|:--------------:|
> |                       |     F1↑        |     AUC↑     |     IoU↑     |     F1↑      |     AUC↑     |     IoU↑     |     F1↑      |     AUC↑     |     IoU↑     |     F1↑      |     AUC↑     |     IoU↑     |     F1↑      |     AUC↑     |     IoU↑     |
> |     Splicing          |     0.975      |     0.991    |     0.963    |     0.984    |     0.994    |     0.965    |     0.985    |     0.993    |     0.964    |     0.984    |     0.994    |     0.963    |     0.982    |     0.995    |     0.965    |
> |     Copy-and-paste    |     0.978      |     0.990    |     0.958    |     0.976    |     0.986    |     0.959    |     0.980    |     0.989    |     0.961    |     0.978    |     0.987    |     0.957    |     0.976    |     0.991    |     0.960    |
> |     Removal           |     0.980      |     0.994    |     0.965    |     0.982    |     0.989    |     0.966    |     0.978    |     0.990    |     0.962    |     0.981    |     0.993    |     0.964    |     0.984    |     0.988    |     0.968    |
> |     Inpainting        |     0.987      |     0.997    |     0.962    |     0.982    |     0.995    |     0.954    |     0.977    |     0.996    |     0.953    |     0.981    |     0.998    |     0.968    |     0.974    |     0.998    |     0.951    |
>
>
> > **Q4**: The paper mentions that random splicing is applied after MPW-VAE. Could the authors clarify why only splicing is considered? What would be the effect of applying other manipulation types, such as random copy-move or random removal, in this step?
>
> **A4**: We adopt random splicing as our manipulation strategy during training because it offers a simple yet effective means for simulating a wide range of real-world tampering patterns, while preserving the self-supervised nature of our framework.
>
> Specifically, random splicing combines a watermarked and a non-watermarked version of the same image using randomly generated binary masks. This technique produces tampered samples that are visually indistinguishable and structurally coherent, both globally and locally. As a result, the model can learn subtle forensic cues without the need for manual annotations.
>
> Furthermore, by splicing the watermarked and non-watermarked versions of the same image, we ensure semantic alignment between the image regions. This is particularly challenging to achieve with techniques like copy-move or removal, since these tampering can often introduce unnatural boundaries or structural inconsistencies, especially in complex scenes. Our strategy enables the model to learn from more complex and challenging examples.
>
> Crucially, by training on spliced images that involve arbitrary pixel-level replacement, our model naturally generalizes to a broad range of tampering types that involve pixel-level modifications at test time, including but not limited to copy-move, random object removal, and inpainting. This generalization capability has been empirically validated through extensive experiments on both synthetic manipulations and real-world tampering datasets, presented in the main paper (e.g., *Section 4.3* in the main paper and *Section F* in the appendix).
>
> We will include these clarifications regarding our training manipulation strategy in the revision.
>
> > **Q5**: What is the approximate inference time of the proposed model for processing a single image, including both watermark verification and manipulation localization?
>
> **A5**: We have conducted additional experiments to measure the inference latency and memory usage of our framework. The detailed performance metrics are summarized in the table below. Specifically, we evaluated the inference time and memory of both our MPW-VAE and the baseline Stable Diffusion VAE (evaluated at 512 × 512), as well as the runtime performance of the MoE-GFN forensic network under the same setting. The results show that MPW-VAE introduces negligible additional computational overhead compared to the original diffusion VAE. Furthermore, the MoE-GFN model exhibits favorable runtime efficiency, with both inference latency and memory consumption remaining within acceptable ranges for practical deployment scenarios.
>
>  We will include these results in the revision.
>
> Table 1. Statistics of runtime latency and VRAM on the AIGC tampering dataset.
> |     Metric           |     Vanilla VAE    |     MPW-VAE       |     MoE-GFN      |
> |----------------|--------------------|-------------------|------------------|
> |     Latency    |     12.69 ms       |     14.99 ms      |     91.88 ms     |
> |     VRAM       |     1681.93 MB     |     2033.97 MB    |     428.32 MB    |
>
>  [1] *Zhai Y, et al. Towards generic image manipulation detection with weakly-supervised self-consistency learning, ICCV 2023.* \
>  [2] *Zhang Z, et al. Image Manipulation Detection with Implicit Neural Representation and Limited Supervision, ECCV 2024.* \
>  [3] *Zhang X, et al. Editguard: Versatile image watermarking for tamper localization and copyright protection,  CVPR 2024.* \
>  [4] *Zhang X, et al. Omniguard: Hybrid manipulation localization via augmented versatile deep image watermarking, CVPR 2025.* \
>  [5] *Sander T, et al. Watermark Anything With Localized Messages, ICLR 2025.*

---

> > ### Comment · Reviewer_9vPj · 2025-08-04
> >
> > Thank you to the authors for their rebuttal. They have addressed nearly all of my concerns. I just have a minor question regarding Table A3: What is the difference between "removal" and "inpainting"? As I understand it, the "removal" manipulation typically involves both deleting an object and filling in the region to produce a natural-looking manipulated image.

---

> > > ### Author Response · Authors · 2025-08-05
> > >
> > > Thank you for your constructive suggestions, which have significantly contributed to the improvement of our work. As you pointed out, the removal operation involves **removing objects** from an image while preserving a natural and continuous background. In contrast, when constructing our dataset, the inpainting operation involves **adding random objects** to a blank background, ensuring that these objects blend seamlessly with the existing scene.
> > >
> > > The key distinction between the two operations lies in their effect on the image: *removal eliminates objects, while inpainting introduces new, randomly chosen elements*. However, both processes share the common goal of maintaining a coherent and natural background. We hope this clarifies your query.

---

> > > > ### Comment · Reviewer_9vPj · 2025-08-05
> > > >
> > > > Thank you for the authors’ clarification. I have no further questions.

---

### Official Review · Reviewer_DW9k · 2025-07-02

**Clarity:** 3
**Significance:** 2
**Originality:** 3
**Rating:** 4
**Confidence:** 5

**Summary:**

The paper introduces StableGuard, a unified diffusion-native framework designed to both embed imperceptible binary watermarks during latent diffusion model (LDM) generation and later verify ownership and localise tampered regions in potentially manipulated images.The core component, Multiplexing-Watermark VAE (MPW-VAE), leverages lightweight residual “watermark adapters” inserted after each decoder block. By toggling adapters on or off, the same latent code yields visually identical watermarked and clean pairs, enabling fully self-supervised training. A Mixture-of-Experts Guided Forensic Network (MoE-GFN) combines three specialised branches—watermark extraction, tamper localisation, and boundary refinement—via a dynamic soft routing mechanism. The entire system is trained end-to-end with a combined similarity, watermark, and tamper loss, so watermark embedding and forensic signal learning reinforce each other. Comprehensive experiments on COCO, a 10k text-to-image dataset, and five tampering benchmarks demonstrate higher watermark bit-accuracy, improved fidelity, and substantially stronger localisation compared to post-hoc watermarking baselines and state-of-the-art passive detectors (MVSS-Net, HDF-Net). Ablation studies confirm the contributions of each module.

**Questions:**

1.How does StableGuard perform under severe downsampling (e.g., 1024→256 px) or aggressive social-media recompression (JPEG Q=30, WebP, HEIC)? Including such experiments would significantly strengthen the practical relevance of your results.

2.For real-world deployment, what are the actual latency and VRAM requirements when generating 1024×1024 images with MPW-VAE, compared to vanilla Stable Diffusion? A table with runtime per image on a modern GPU (e.g., RTX 4090) would be highly informative.

3.Your training masks are random binary regions, whereas real tampering often aligns with semantic boundaries. Have you experimented with using SAM-based masks or object-aware masks to narrow this domain gap?

**Ethical Concerns:**

["NO or VERY MINOR ethics concerns only"]

**Final Justification:**

Based on previous comments and the author's response, I keep my score.

**Limitations:**

Yes.

**Paper Formatting Concerns:**

1.Figure 2(b) labels are too small to read at print scale; consider enlarging the font size.

**Quality:**

3

**Strengths And Weaknesses:**

This paper demonstrates several notable strengths.
1.The paper presents a well-designed approach with clear algorithmic formulation, sensible loss functions, and comprehensive ablation studies. The authors thoroughly evaluate the impact of MPW-VAE variants, expert configurations, and routing strategies, which strengthens the credibility of the results.

2.Experiments cover multiple degradation scenarios (noise, JPEG compression, Poisson noise), demonstrating that the method remains effective under moderate distortions.

3.The work addresses an important and timely problem: provenance verification and tamper localisation for AI-generated images. The results show that end-to-end joint training of watermark embedding and localisation can achieve substantial gains over existing post-hoc methods.

However, there are also important weaknesses to consider.
1.All localisation benchmarks are synthetic and based on LDM-generated edits; there is no evaluation on human-made forgeries or realistic manipulations such as colour grading, rescaling, or recompression commonly seen in the wild.

2.Computational costs are reported only in FLOPs, without details on actual runtime latency or memory footprint during inference, making it difficult to assess feasibility for deployment.

3.The experiments do not address resilience to aggressive real-world compression, and the false-positive rate on clean, non-watermarked images remains unexplored, which raises concerns about practical reliability.

---

> ### Author Rebuttal · Authors · 2025-07-28
>
> Thank you for your insightful comments and valuable suggestions. We have revised our paper based on your feedback. Here are our responses to your comments:
>
> > **Q1**: All localisation benchmarks are synthetic and based on LDM-generated edits; there is no evaluation on human-made forgeries or realistic manipulations such as colour grading, rescaling, or recompression commonly seen in the wild.
>
>  **A1**: We believe there might be a misunderstanding. We have indeed conducted comparison experiments on five standard tampering localization datasets involving human-made forgeries: CASIA, NIST16, Columbia, Coverage, and IMD20, as detailed in *Appendix Section F*. These datasets cover various manipulations, including splicing, copy-move, object removal, etc. The reported results demonstrate the superior performance and practical effectiveness of our method in handling real-world forgery scenarios.
>
>  For experimental results on real-world image degradation, such as color transformations and aggressive compression, please refer to **A4**.
>
> > **Q2**: Computational costs are reported only in FLOPs, without details on actual runtime latency or memory footprint during inference, making it difficult to assess feasibility for deployment.
>
>  **A2**: Thank you for your suggestion. We run additional experiments to assess inference latency and VRAM usage on an RTX 4090 GPU for our MPW-VAE, the vanilla Stable Diffusion VAE, and the MoE-GFN forensic network (evaluated at 512×512). Results are in Table 1.
>
> Our findings show that MPW-VAE adds negligible overhead compared to the vanilla VAE in latency and VRAM, while the MoE-GFN model performs efficiently, making the framework viable for practical forensic deployment.
>
> Table 1. Statistics of runtime latency and VRAM on the AIGC tampering dataset.
> |   Metric             |     Vanilla VAE    |     MPW-VAE       |     MoE-GFN      |
> |----------------|--------------------|-------------------|------------------|
> |     Latency    |     12.69 ms       |     14.99 ms      |     91.88 ms     |
> |     VRAM       |     1681.93 MB     |     2033.97 MB    |     428.32 MB    |
>
>
> > **Q3**: The experiments do not address resilience to aggressive real-world compression, and the false-positive rate on clean, non-watermarked images remains unexplored, which raises concerns about practical reliability.
>
>  **A3**: For robustness to aggressive compression, please refer to **A4**, where we provide comprehensive evaluations under various degradations.
>
> As for the concern about false positives on clean, non-watermarked images, we would like to clarify that our method is specifically designed as a proactive watermarking framework [1,2,3], in which the watermark serves as a critical bridge to achieve copyright protection and tampering localization simultaneously. Unlike passive detection methods, our approach assumes that images are proactively embedded with a watermark at creation time. As such, clean images without watermarks fall outside the operational scope of our system and can be conservatively interpreted as “fully tampered”.
>
> Given this design, evaluating false positives on truly non-watermarked images is not applicable. Instead, we assess practical reliability by measuring false positive detections in (1) untampered regions of partially tampered watermarked images and (2) fully untampered, watermarked images. As shown in Table 2, our method achieves the lowest false positive rate (defined as the ratio of false positive pixels to the total untampered pixel count) in both cases, demonstrating its effectiveness and reliability.
>
> We will include these results and explanations in the revised manuscript to clarify the motivation and scope of our framework.
>
> Table 2. Comparison of false positive rate (FPR ↓) on the AIGC tampering dataset.
> |     Type             |     EditGuard    |     OmniGuard    |     WAM        |     Ours       |
> |---------------------|------------------|------------------|----------------|----------------|
> |     Partially tampered        |     0.0422       |     0.0191       |     0.0045     |     **0.0023**     |
> |     Fully untampered      |     0.0028       |     0.0024       |     0.0019     |     **0.0016**     |
>
> > **Q4**: How does StableGuard perform under severe downsampling (e.g., 1024→256 px) or aggressive social-media recompression (JPEG Q=30, WebP, HEIC)? Including such experiments would significantly strengthen the practical relevance of your results.
>
>  **A4**: We have conducted additional experiments involving a diverse set of image degradation operations, representative of real-world scenarios. These include: (1) aggressive image compression, (2) severe resolution downsampling (1024→256 px is equivalent to a rescale factor of 0.25), and (3) color transformations. The specific parameter ranges for each transformation are summarized in the table below.
>
> As shown in the results, our method maintains high watermark extraction accuracy and exhibits reasonable tampering localization performance. This behavior is consistent with our design intuition: watermark extraction relies on global cues, which are more robust to distortions, while tampering localization depends on local consistency, and is naturally more sensitive to compression and resampling artifacts. Nevertheless, the degradation in localization performance remains moderate and within acceptable bounds. These results confirm the real-world applicability and resilience of our approach.
>
> We will incorporate these results in the revised version.
>
> **Table 3**. Performance under various degradations on AIGC tampering dataset.
> |     Type                 |     EditGuard |              |     OmniGuard    |              |        WAM      |              |       Ours      |           |
> |----------------------|:-------------:|:------------:|:----------------:|:------------:|:---------------:|:------------:|:---------------:|:---------:|
> |                      |    Bit Acc.↑   |      F1↑      |      Bit Acc.↑    |       F1↑     |     Bit Acc.↑    |       F1↑     |     Bit Acc.↑    |     F1↑    |
> |     Clean            |       99.78   |     0.938    |       98.11      |     0.863    |       98.17     |     0.920    |    **99.98**    | **0.972** |
> |     JPEG (Q=30)      |       62.30   |     0.230    |       63.34      |     0.311    |       80.96     |     0.532    |    **98.87**    | **0.866** |
> |     WebP (Q=50)      |       52.88   |     0.233    |       56.16      |     0.307    |       84.87     |     0.538    |    **98.95**    | **0.703** |
> |     HEIC (Q=50)      |       66.53   |     0.335    |       59.41      |     0.323    |       87.46     |     0.402    |    **98.36**    | **0.748** |
> |     Resale (0.25)    |       41.77   |     0.256    |       54.95      |     0.343    |       95.98     |     0.821    |    **97.55**    | **0.858** |
> | Brightness (0.8-1.2) |     91.35     |     0.504    |       89.55      |     0.741    |       94.71     |     0.816    |    **98.94**    | **0.860** |
> | Contrast (0.8-1.2)   |     90.84     |     0.788    |       91.45      |     0.836    |       95.00     |     0.824    |    **97.93**    | **0.836** |
> | Saturation (0.8-1.2) |     92.93     |     0.817    |       93.44      |     0.856    |       95.69     |     0.814    |    **97.98**    | **0.911** |
>
> > **Q5**: For real-world deployment, what are the actual latency and VRAM requirements when generating 1024×1024 images with MPW-VAE, compared to vanilla Stable Diffusion? A table with runtime per image on a modern GPU (e.g., RTX 4090) would be highly informative.
>
>  **A5**: We compared inference latency and VRAM usage of our MPW-VAE with the vanilla Stable Diffusion VAE for generating 1024×1024 images in Table 4. Our MPW-VAE shows similar runtime and memory consumption to the original VAE, with only minimal overhead, proving its efficiency for practical deployment.
>
> Table 4. Latency and VRAM usage of MPW-VAE vs. vanilla Stable Diffusion VAE.
> |     Metric    |     Vanilla VAE    |     MPW-VAE       |
> |----------------|--------------------|-------------------|
> |     Latency    |     127.03 ms      |     169.40 ms     |
> |     VRAM       |     4598.30 MB     |     6006.15 MB    |
>
> >**Q6**: Your training masks are random binary regions, whereas real tampering often aligns with semantic boundaries. Have you experimented with using SAM-based masks or object-aware masks to narrow this domain gap?
>
>  **A6**: Thank you for your careful comment. In our implementation, we employed a hybrid masking strategy that combines both random binary masks and semantic-aware masks to simulate diverse tampering patterns. Specifically, with 50% probability, we apply random binary masks, while for the remaining 50%, we utilize segmentation maps extracted by SAM. These SAM-based masks are randomly drawn from a pre-collected pool of segmentation shapes and are not image-paired, allowing us to inject shape diversity without requiring additional supervision.
>
> However, in the original manuscript, we referred to this mask generation process in a simplified manner as “random binary masks”, which may have caused confusion. We will revise the manuscript to clearly reflect the actual hybrid masking strategy, thereby providing a more accurate description of our method. We will also release our code, pre-trained models, and datasets to facilitate reproducibility and foster future research upon acceptance.
>
>
> > **Q7**: Figure 2(b) labels are too small to read at print scale; consider enlarging the font size.
>
>  **A7**: We will revise this in the revision.
>
>  [1] *Zhang X, et al. Editguard: Versatile image watermarking for tamper localization and copyright protection,  CVPR 2024.* \
>  [2] *Zhang X, et al. Omniguard: Hybrid manipulation localization via augmented versatile deep image watermarking, CVPR 2025.* \
>  [3] *Sander T, et al. Watermark Anything With Localized Messages, ICLR 2025.*

---

> > ### Comment · Reviewer_DW9k · 2025-08-07
> >
> > Thanks all authors for the rebuttal. The response addressed most of my concerns, I have no further questions.

---

> > > ### Author Response · Authors · 2025-08-07
> > >
> > > We sincerely thank you for your review and comments which have improved our paper.

---

### Official Review · Reviewer_Fd7Y · 2025-07-15

**Clarity:** 4
**Significance:** 4
**Originality:** 4
**Rating:** 5
**Confidence:** 5

**Summary:**

The work proposes a novel framework StableGuard to address the need for robust copyright protection and tampering detection in images generated by LDMs. The framework consists of two components: Multiplex Watermark VAE (MPW-VAE) and Mixture-of-Experts Guided Forensic Network (MoE-GFN) which are optimized jointly in a self-supervised fashion. MPW-VAE integrates watermark via residual-based adapters which are inserted after each VAE decoder block. MoE-GFN consists of three blocks: Watermark extraction, Tampering localization and Boundary enhancement and for all three of these, a dynamic soft router is employed to adapt task relevant features. StableGuard is trained on COCO training set and evaluated on COCO test set and custom T2I dataset. It is evaluated for both watermark extraction and tampering localization tasks.

**Questions:**

- For evaluation of watermark extraction, only quantitative comparison has been shown in Table 1. Figure 3 only shows the qualitative results of the proposed solution, it would be good to see the qualitative comparison with the existing methods also to access the visual quality of the watermarked images for all methods.
- For quantitative comparisons, one of the recent work is missing:
a. Zhang, X., Tang, Z., Xu, Z., Li, R., Xu, Y., Chen, B., Gao, F. and Zhang, J., 2025. Omniguard: Hybrid manipulation localization via augmented versatile deep image watermarking. In Proceedings of the Computer Vision and Pattern Recognition Conference (pp. 3008-3018).
- For WEE, TLE, BEE more discussion is required for each component contribution, in ablation study 3rd, 4th and 5th rows marks very minor difference for F1, AUC, IoU and Bit accuracy with respect to each other. It raises the question of why three different extraction methods were required in the first place?
- Details of custom T2I dataset used for evaluation of watermark extraction are limited, as this is custom it would be beneficial to see the stats of the prompts used for this dataset generation like diversity of this dataset, which categories were used for selection, how were these selected?
- Eq.(8) mentions LPIPS as Similarity loss, I think the correct version of mentioning LPIPS here is perceptual loss as we are defining the loss function and LPIPS is mostly used as an evaluation term.

**Ethical Concerns:**

["NO or VERY MINOR ethics concerns only"]

**Final Justification:**

All of the concerns have been addressed by the authors, and comparison results are also shown showing the effectiveness of the proposed method.

**Paper Formatting Concerns:**

- Section 3.5 explains the loss functions of the proposed framework as a different section, as Section 3.3 and 3.4 already discusses the main components of the proposed framework, it would be more readable if 3.5 could be integrated within 3.3 and 3.4

**Quality:**

4

**Strengths And Weaknesses:**

Strengths:
+ The paper is well-written and easy to understand.
+ The idea of joint optimization of watermark integration and tampering localization is novel and interesting.
+ Evaluations show state-of-the-art results in forensic accuracy and robustness.

---

> ### Author Rebuttal · Authors · 2025-07-27
>
> Thank you for your insightful comments and valuable suggestions. We have revised our paper based on your feedback. Here are our responses to your comments:
>
> > **Q1**: For evaluation of watermark extraction, only quantitative comparison has been shown in Table 1. Figure 3 only shows the qualitative results of the proposed solution, it would be good to see the qualitative comparison with the existing methods also to access the visual quality of the watermarked images for all methods.
>
>  **A1**: Thank you for the constructive suggestion. Due to the text-only policy of the rebuttal, we are unable to include visualizations at this stage. We will incorporate these results in the revision.
>
> > **Q2**: For quantitative comparisons, one of the recent work is missing: a. Zhang, X., Tang, Z., Xu, Z., Li, R., Xu, Y., Chen, B., Gao, F. and Zhang, J., 2025. Omniguard: Hybrid manipulation localization via augmented versatile deep image watermarking. In Proceedings of the Computer Vision and Pattern Recognition Conference (pp. 3008-3018).
>
>  **A2**: We appreciate the reviewer’s suggestion to include OmniGuard [1] for a more comprehensive comparison. We have conducted additional experiments and incorporated quantitative results for OmniGuard in Tables 1 and 2 below. Our method consistently outperforms OmniGuard in both watermark extraction fidelity and tampering localization accuracy. These results, along with a detailed visual comparison, will be included in the revised version of the paper.
>
>
> **Table 1**: Quantitative comparison on watermarking performance on COCO and the T2I dataset.
> |     Method       |     B.L.    |     COCO     |              |              |             |                 |     T2I     |              |              |             |                 |
> |------------------|-------------|---------------------|--------------|--------------|-------------|-----------------|--------------------|--------------|--------------|-------------|-----------------|
> |                  |             |     PSNR↑            |     SSIM↑     |     LPIPS↓    |     FID↓     |     Bit Acc.↑    |     PSNR↑           |     SSIM↑     |     LPIPS↓    |     FID↓     |     Bit Acc.↑    |
> |     OmniGuard    |     100     |     37.54           |     0.950    |     0.072    |     20.1    |     98.11       |     37.32          |     0.944    |     0.070    |     20.0    |     98.09       |
> |     Ours         |     128     |     **40.10**       |     **0.966** |     **0.070**    |     **19.9**    |     **99.87**       |     **40.11**          |     **0.968**    |     **0.069**    |     **19.8**    |     **99.88**       |
>
> **Table 2**: Localization precision comparison on the AIGC tampering dataset.
> |     Method       |     SD-Inp.    |              |              |     SD-XL    |              |              |     Kand.    |              |              |     Cont.    |              |              |     LAMA     |              |              |
> |------------------|:----------------------:|:--------------:|:--------------:|:--------------:|:--------------:|:--------------:|:------------------:|:--------------:|:--------------:|:-------------------:|:--------------:|:--------------:|:--------------:|:--------------:|:--------------:|
> |                  |     F1↑               |     AUC↑      |     IoU↑      |     F1↑       |     AUC↑      |     IoU↑      |     F1↑           |     AUC↑      |     IoU↑      |     F1↑            |     AUC↑      |     IoU↑      |     F1↑       |     AUC↑      |     IoU↑      |
> |     OmniGuard    |     0.853            |     0.964    |     0.810    |     0.867    |     0.973    |     0.824    |     0.868        |     0.966    |     0.830    |     0.858         |     0.965    |     0.815    |     0.864    |     0.969    |     0.823    |
> |     Ours         |     **0.980**            |     **0.993**    |     **0.962**    |     **0.981**    |     **0.991**    |     **0.961**    |     **0.980**        |     **0.992**    |     **0.960**    |     **0.981**         |     **0.993**    |     **0.963**    |     **0.979**    |     **0.993**    |     **0.961**    |
>
> > **Q3**: For WEE, TLE, BEE more discussion is required for each component contribution, in ablation study 3rd, 4th and 5th rows marks very minor difference for F1, AUC, IoU and Bit accuracy with respect to each other. It raises the question of why three different extraction methods were required in the first place?
>
>  **A3**: Each expert in the proposed MoFE module is designed to extract a specific and complementary forensic signal: the Watermark Extraction Expert (WEE) captures global watermark patterns, the Tampering Localization Expert (TLE) focuses on localized inconsistencies, and the Boundary Enhancement Expert (BEE) enhances sensitivity to structural boundaries indicative of manipulation. Although the individual removal of these experts (rows 3–5 in Table 4) results in only modest drops in F1, AUC, IoU, and Bit Accuracy, this is primarily due to the model operating near a performance ceiling, where marginal gains from each expert appear limited in isolation.
>
> However, their contributions are complementary rather than redundant. The significantly larger performance drop observed when the entire MoFE module is removed (row 2) clearly demonstrates their combined effectiveness. This indicates that while each expert alone contributes moderately, their integration is crucial for the model’s robustness and accuracy.
>
> We will revise the manuscript to make this design rationale and its empirical support more explicit, including further discussion of the roles of each expert and their synergistic effect (as illustrated in Figure 5).
>
> > **Q4**: Details of custom T2I dataset used for evaluation of watermark extraction are limited, as this is custom it would be beneficial to see the stats of the prompts used for this dataset generation like diversity of this dataset, which categories were used for selection, how were these selected?
>
>  **A4**: To ensure semantic diversity and broad coverage of visual concepts in our T2I dataset, we curated prompts from five distinct thematic categories: natural landscapes, urban cityscapes, indoor scenes, animals, and human-centric scenes. For each category, we constructed 200 unique prompts, resulting in 1,000 diverse text inputs spanning a wide range of visual attributes. Specifically, to balance structural consistency with semantic variability, these prompts were generated by combining curated templates with randomized keyword insertions from controlled vocabularies. Each prompt was used to synthesize 10 images, yielding 10,000 generated samples in total.
>
>  We will include a detailed summary of prompt statistics, selection methodology, and representative examples in the revised version, and will release the full dataset (including prompts, original images, and tampering images) to facilitate future research.
>
> > **Q5**: Eq. (8) mentions LPIPS as Similarity loss, I think the correct version of mentioning LPIPS here is perceptual loss as we are defining the loss function and LPIPS is mostly used as an evaluation term.
>
>  **A5**: We will fix this in the revision.
>
> > **Q6**: Section 3.5 explains the loss functions of the proposed framework as a different section, as Section 3.3 and 3.4 already discusses the main components of the proposed framework, it would be more readable if 3.5 could be integrated within 3.3 and 3.4
>
>  **A6**: We will revise this accordingly.
>
> [1] *Zhang X, et al. Omniguard: Hybrid manipulation localization via augmented versatile deep image watermarking, CVPR 2025.*

---

### Note · Authors · 2025-08-11

We sincerely thank the AC and reviewers for their thoughtful and constructive feedback.

In the original submission, the reviewers recognized several key strengths of our work:

1. **Novelty**. The joint optimization of watermark integration and tampering localization, combined with the framework design of MPW-VAE and MoE-GFN, introduces a new and distinctive approach.
2. **Importance**. The work addresses the critical challenge of provenance verification and tamper localization for LDM-generated images, offering a valuable contribution to the field.
3. **Clarity**. The paper is clearly written, well structured, and easy to follow.
4. **Effectiveness**. Extensive experiments demonstrate that StableGuard achieves strong robustness against diverse tampering scenarios and enables accurate watermark extraction on both LDM-based and real-world data.

The main points raised for further evaluation were:

1. Comparison with two recent baselines, OmniGuard and WAM.
2. Assessment of computational efficiency.
3. Robustness under a wider range of image degradation scenarios.
4. Robustness under varying tampering rates.
5. Dataset construction and detection accuracy under different tampering types.

These points **focus on expanding the empirical validation of StableGuard’s applicability, rather than questioning its core novelty, contributions, or effectiveness**. All have been thoroughly addressed during the rebuttal period, with additional experiments and clarifications provided. **Reviewers have acknowledged these improvements.**

We believe the revised work delivers a **novel, impactful, and rigorously validated** solution to robust watermarking and tampering localization.

---

### Decision · Program_Chairs · 2025-09-17

**Decision:**

Accept (poster)

**Comment:**

The paper proposes a framework for copyright protection and tampering localization in images generated by LDMs. All four reviewers gave positive scores: two borderline accepts and two accepts. The major concerns raised by the reviewers include additional comparisons with two baselines and experimental analysis on the robustness and efficiency. The authors addressed all the concerns during the discussion period. Based on all of these, the decision is to recommend the paper for acceptance.